# Characterizing the near-global cloud vertical structures over land using high-resolution radiosonde measurements

Hui Xu[1], Jianping Guo[1], Bing Tong[2], Jinqiang Zhang[3], Tianmeng Chen[1], Xiaoran Guo[1], Jian Zhang[4], and Wenqing Chen[5]

[1]State Key Laboratory of Severe Weather, Chinese Academy of Meteorological Sciences, Beijing, 100081, China
[2]State Key Laboratory of Urban and Regional Ecology, Research Center for Eco-environmental Sciences, Chinese Academy of Sciences, Beijing, 100085, China
[3]Key Laboratory of Middle Atmosphere and Global Environment Observation, Institute of Atmospheric Physics, Chinese Academy of Sciences, Beijing, 100029, China
[4]Hubei Subsurface Multi-Scale Imaging Key Laboratory, Institute of Geophysics and Geomatics, China University of Geosciences, Wuhan, 430074, China
[5]Business Science and Technology Division, National Meteorological Information Centre, Beijing, 100081, China

*Correspondence to*: Jianping Guo (jpguocams@gmail.com) and Bing Tong (bingtong@rcees.ac.cn)

**Abstract.** Cloud remains one of the largest uncertainties in weather and climate research due to the lack of fine-resolution observations of cloud vertical structure (CVS) on large scale. In this study, near-global CVS is characterized by high-vertical-resolution twice daily radiosonde observations from 374 stations over land, which distributed in Europe, North America, East Asia, Austria, Pacific Ocean, and Antarctica. To this end, we initially develop a novel method to determine CVS, by combining both the vertical gradients of air temperature and relative humidity (RH) and the altitude-dependent thresholds of RH. It is found that the cloud base heights (CBHs) from the radiosondes have a higher correlation coefficient (R = 0.91) with the CBHs from millimeter wavelength cloud radar than those from the ERA5 reanalysis (R = 0.49). Overall, cloudy skies occur 65.3 % (69.5 %) of the time, of which 55.4 % (53.8 %) are one-layer clouds at 0000 (1200) UTC. Most multi-layer clouds are two-layer clouds, accounting for 62.2 % (61.1 %) among multi-layer clouds for 0000 (1200) UTC. Geographically, one-layer clouds tend to occur over arid regions, whereas two-layer clouds do not show any clear spatial preference. The cloud bases and tops over arid regions are higher compared with humid regions albeit with smaller cloud thickness (CT). Clouds tend to have lower bases and thinner layer thicknesses as the number of cloud layer increases. The global mean CT, CBH, and cloud top height (CTH) are 4.89 ± 1.36 (5.37 ± 1.58), 3.15 ± 1.15 (3.07 ± 1.06), and 8.04 ± 1.60 (8.44 ± 1.52) km above ground level (AGL) at 0000 (1200) UTC, respectively. The occurrence frequency of clouds is bimodal, with lower peaks between 0.5 and 3 km AGL and upper peaks between 6 and 10 km AGL. The CBH, CTH and CT undergo almost the same seasonality, namely, their magnitudes in boreal summer are greater than in boreal winter. As expected, the occurrence frequencies of clouds exhibit pronounced diurnal cycles in different seasons. In boreal summer, clouds tend to form as sun rises and the occurrence frequencies increase from morning to later afternoon, with the peak in the early afternoon at the altitude of 6–12 km AGL; while in boreal winter, clouds have peak occurrence frequencies in the morning. The relations between surface meteorological variables and moisture with CBH are investigated as well, showing that CBH are generally more significantly correlated with

2m relative humidity ($RH_{2m}$) and 2m air temperature ($T_{2m}$) than with surface pressure and 10m wind speed. Larger $T_{2m}$ and smaller $RH_{2m}$ always correspond to higher CBH. In most cases CBHs are negatively correlated to soil water content. The near-global CVS obtained from high-vertical-resolution radiosonde in this study can provide key data support for improving the accuracy of cloud radiative forcing simulation in climate models.

## 1 Introduction

Clouds cover nearly two-thirds of Earth's surface area and have a significant impact on the radiative budget of Earth-atmosphere system (e.g., Ramanathan et al., 1989; Houghton et al., 1996; Crewell et al., 2004; Stephens et al., 2005; Trenberth et al., 2009). Many studies have pointed out that the cloud radiative effects (CRE) are critically dependent on the height of clouds (e.g., Wielicki et al. 1995; Weare, 2000; Nam et al., 2012; Stephens et al., 2012; Wild, 2012; Lee et al., 2015; George et al., 2018). Different cloud types defined by cloud height can even cause two opposite CREs (i.e., the "greenhouse" and the

"umbrella" effects, Meehl and Washington, 1995). Low and thick clouds tend to cool the surface through reflecting solar radiation, whereas high and thin cirrus clouds tend to warm the surface by preventing longwave radiation emitting outward (Liou, 1986; Naud et al., 2003; Solomon et al., 2007). Even small variations in cloud vertical structure (CVS) can lead to significant differences in the mean CRE and radiative heating/cooling rates (Costa-Surós et al., 2014). However, the vertical structures of clouds are often not accurately represented in current climate models, leading to large uncertainties in estimating

the CRE (e.g., Randall et al., 2003; Waliser et al., 2009; Cesana and Chepfer, 2012; Cesana and Waliser, 2016; IPCC, 2021). Therefore, it is imperative to procure high-quality global CVS to improve the predictive capabilities of current climate models.

      Satellite observations are efficient in detecting cloud properties. Passive satellite sensors like moderate resolution imaging spectroradiometer (MODIS) can resolve a global coverage of cloud fraction and top height, but not the vertical information of clouds (Platnick et al., 2003). Chang and Li (2005) retrieved near-global CVS for one-layer and overlapped clouds based on

MODIS data. However, these retrievals lack the vertical structures of three- or more-layer clouds. Active sensors such as the cloud-aerosol lidar with orthogonal polarization (CALIOP) onboard CALIPSO (Winker et al., 2009) and the millimeter wavelength cloud profiling radar (CPR) onboard CloudSat (Stephens et al., 2002) can provide vertical structure of cloud layers on global scale (Oreopoulos et al., 2017). However, active sensors have relatively long revisit periods (e.g., 16-day) and narrow nadir views (e.g., Winker et al., 2007; Kim et al., 2011; Guo et al., 2016).

Ground-based instruments, such as lidars (Gouveia et al., 2017), ceilometers (Costa-Surós et al., 2013), and cloud radars (Mace et al., 1998), have proven to be effective in providing CVS with continuous temporal coverage and relatively high accuracy (Hahn et al., 2001; Zhou et al., 2020). Lidars and ceilometers can pinpoint cloud bases and cloud radars can resolve multi-layer clouds (Willen et al., 2005; Nowak et al., 2008; Reddy et al., 2018). The US Department of Energy's Atmospheric Radiation Measurement Program (ARM) Climate Research Facility (http://www.arm.gov) provides ground-based radar and

lidar observations at fixed field sites: North Slope of Alaska (NSA; Zhang et al., 2017), Southern Great Plains (SGP; Dong et al., 2008), Tropical Western Pacific (TWP; Comstock et al., 2013), and Eastern North Atlantic (ENA; Giangrande et al., 2019),

and several mobile field sites (Cadeddu et al., 2013). These measurements provide information on the vertical structure of clouds (Stokes and Schwartz, 1994; Ackerman and Stokes, 2003), and have been widely used to study the cloud properties on global climate (Mace and Benson, 2008; Chandra et al., 2015). However, the global coverage of these instruments is too sparse and limited.

Radiosonde can provide reliable measurements of the profiles of air temperature (T) and relative humidity (RH) at a great many locations on a global scale (although underrepresented over oceans), making it possible to obtain global CVS given that the cloud formation is highly associated with the water vapor and thermal conditions of the atmosphere (e.g., Poore et al., 1995; Wang and Rossow, 1995; Chernykh and Eskridge, 1996; Wang et al., 2000; Minnis et al., 2005; Zhang et al., 2010). In order to detect CVS from radiosonde observations, two main methods have been proposed. One is the threshold method, in which cloud layers are determined using RH thresholds. Poore et al. (1995) proposed a T-dependent dewpoint depression threshold for cloud detection, and they found that only high clouds exhibited strong latitudinal and seasonal variation in the thickness of cloud layer. Wang and Rossow (1995) detected cloud layers using single RH threshold, with the maximum and minimum RH thresholds of 87 % and 84 %, respectively. They demonstrated that the occurrence frequency of multi-layer clouds varied geographically and multi-layer clouds occurred most frequently in the tropics. Zhang et al. (2010) improved the single threshold method by using an altitude-dependent RH thresholds to characterize the base and top of cloud layers, and they demonstrated that multilayer clouds occurred more frequently in the summer. Another method is the gradient method, in which cloud layers are obtained by examining the variations of RH and T profiles. Chernykh and Eskridge (1996) used a second derivative of the vertical profiles of RH and T to determine cloud boundaries, and they indicated that the accuracy of the prediction of cloud level is independent of the level type and location. Dzambo and Turner (2016) identified cirrus based on a cirrus cloud detection algorithm by using radiosonde and cloud radar data and found that RH with respect to ice within cirrus clouds varied seasonally, with maximum occurring in winter and minimum in summer. To ensure the radiosonde measurements were collocated with the appropriate MMCR measurements, they established temporal ("lag time") and spatial restrictions. However, by comparing the CVS derived from above methods with ground-based remote sensing measurements, Costa-Surós et al. (2014) indicated that the performances of these mentioned CVS retrieval methods may need further improvement. The possible reasons can be concluded as (1) the vertical resolution of atmospheric profiles provided by radiosonde is low (e.g., 76 meters (m); Poore et al., 1995), and (2) refined RH thresholds remain lacking for cloud detection.

With the emergence of growing number of high-vertical-resolution (5–10 m) radiosonde measurements worldwide, improved retrievals of CVS on large scale are now plausible. However, to our knowledge no studies have reported near-global CVS from high-resolution radiosonde measurements. The main objective of present study is to provide the first attempt to retrieve near-global vertical structures of clouds, including the number of cloud layers, cloud base height (CBH), cloud top height (CTH), and cloud thickness (CT) of each layer, using two years' worth (2018–2019) of high-vertical-resolution (5–10 m) radiosonde observations from 374 radiosonde stations across the world (e.g., Europe, North America, East Asia, Austria, Pacific Ocean, and Antarctica). In order to obtain better CVS results, we first develop a novel CVS detection method that integrates the two main methods mentioned above by considering both the vertical gradients of RH and T, as well as the

altitude-dependent thresholds of RH. The radiosonde derived CVS is compared with that obtained by millimeter wavelength cloud radar (MMCR) observations and ERA5 reanalysis. Unless otherwise noted, CBH represents the base for the lowermost cloud layer, CTH represents the top for the uppermost cloud layer, and CT represents the total cloud thickness (CTH – CBH). The remainder of this paper proceeds as follows: Section 2 describes the data and methods to determine the CVS. The performance of the CVS retrieved in this study is evaluated in Section 3. We analyse the frequency of clouds with various layers, their vertical and horizontal distributions, and the diurnal variation of cloud occurrence frequency with height. To examine the potential key factors that affect the CVS we also investigate the relationship between CBH, surface meteorological variables, and moisture. The main conclusions are summarized in Section 4.

## 2. Data and Methods

### 2.1 Data

#### 2.1.1 High-resolution radiosonde data

Radiosonde instruments provide in situ measurements of atmospheric environmental variables, including RH, T, pressure, and wind profiles, which are recognized to be able to derive the vertical distribution of cloud (Chernykh and Eskridge, 1996). Radiosonde measurements are usually made twice a day at 0000 and 1200 UTC. For high-resolution radiosondes, the sampling period is approximately 1–2 seconds (s), and the vertical resolution is approximately 5–10 m throughout the atmosphere (Guo et al., 2016; 2019; 2021). In this study, high-vertical-resolution radiosonde measurements obtained from 426 sites across the globe are provided by several organizations, including the China Meteorological Administration (CMA), the National Oceanic and Atmospheric Administration (NOAA) of United States, the German Deutscher Wetterdienst (Climate Data Center), the Centre for Environmental Data Analysis (CEDA) of United Kingdom, the Global Climate Observing System (GCOS) Reference Upper Air Network (GRUAN), and University of Wyoming. In total, there are more than 400 profiles at most radiosonde stations, which provide a roughly even distribution near-global and sufficient samples to characterize the climatology of the CVS. To minimize the uncertainty in results, the radiosonde stations with the profile number less than 400 are not included, and radiosonde measurements at the remaining 374 stations are used to derive near-global CVS. The remaining 374 radiosonde stations include 120 L-band radiosonde stations from the CMA, 150 stations from NOAA, and 104 stations from the German Deutscher Wetterdienst (Climate Data Center), the CEDA, the GRUAN, and the University of Wyoming. Overall, 407,688 soundings at near-global 374 high-vertical-resolution radiosonde stations for the period of 2018–2019 are used here, including 206,130 soundings at 0000 UTC and 201,558 soundings at 1200 UTC. The geographic distribution of the numbers of profiles at these radiosonde stations is displayed in Fig. 1.

The Vaisala RS92 radiosonde is widely used by NOAA, the German Deutscher Wetterdienst, the CEDA, and the University of Wyoming. The Vaisala RS92 humidity sensor measures RH every 2 s (Wang et al., 2018), and its uncertainty is 5 % RH (Jauhiainen and Lehmuskero, 2005). Due to solar radiation heating, the RH data results in a dry bias in the upper

troposphere (Vömel et al. 2007). Several correction algorithms have been developed to correct the solar radiation dry bias (e.g., Vömel et al., 2007; Cady-Pereira et al., 2008; Yoneyama et al., 2008; Miloshevich et al., 2009; Wang et al., 2013). The Vaisala RS41 radiosonde is used in the stations of GRUAN. Since temperature of the humidity sensor can be measured by the temperature sensor and taken into account in the RH calculation, no separate solar radiation dry bias correction is needed for the RS41 humidity measurement (Jensen et al., 2016). The RS41 humidity sensor has an uncertainty of 3 % RH (Vaisala, 2017). The GTS1 digital radiosonde is used by CMA, having the advantages of high sensitivity, quick sampling, and small volume (Li, 2006; Bian et al., 2011; Chen et al., 2021). The humidity sensor of GTS1 samples RH at a time interval of approximately 1 s, with the uncertainty about 5 % RH (Li et al., 2009). The specifications for the Vaisala RS92, Vaisala RS41, and GTS1 digital radiosonde are shown in Table 1.

### 2.1.2 Ka-band millimeter-wave cloud radar

The Ka-band (35 GHz) MMCR (Moran et al., 1998) has the advantage of penetrating clouds and continuously detecting the vertical structure of clouds, which can reach an accuracy of 150 m for cloud boundaries (Clothiaux et al., 2000; Hollars et al., 2004). The CVS extracted from MMCR are widely used as reference for inter-comparison with radiosonde derived CVS (e.g., Naud et al., 2005; Zhang et al., 2013). In this study, we use the data measured by a Ka-band MMCR installed at Beijing Nanjiao weather observatory (BNWO, 39.81°N, 116.47°E, 32 m above sea level, ASL) from the Meteorological Observation Center of the CMA during the period of 2019. This MMCR can measure an altitude of 15 km above ground level (AGL) with a vertical resolution of 30 m and a temporal resolution of 1 min. The CBHs and CTHs are identified based on the minimum threshold method by using the MMCR reflectivity measurements (Zhang et al., 2019).

### 2.1.3 ERA5 reanalysis

The ERA5 is the fifth generation of global atmospheric, land and oceanic climate reanalysis produced by the European Centre for Medium-Range Weather Forecasts (Bell et al., 2021). The ERA5 reanalysis dataset, covering 1979 to present, is produced using 4D-Var data assimilation and is closely associated with the excellence of the forecast products. The temporal and spatial resolutions of ERA5 can reach up to 1 hour (h) and 0.25° × 0.25°, respectively (Hersbach et al., 2020). The product consists of single level variables (e.g., CBH and total cloud fraction) on one level or near surface, pressure level variables (e.g., temperature and relative humidity) on 37 levels from 1000 to 1 hPa, and model level variables on 137 levels from the surface to a height of 80 km ASL (https://cds.climate.copernicus.eu/cdsapp#!/dataset/reanalysis-era5-complete?tab=overview). As a key variable that links land surface to cloud formation, soil moisture from the ERA5 reanalysis has been widely used in the analysis of land-atmosphere coupling (Sun et al., 2020). By using the ERA5 reanalysis in East Asia, Wei et al. (2021) explored the relationships between soil moisture, land surface sensible and latent heat fluxes, and CBH, and found the negative correlations between soil moisture and CBH. In this study, we use five meteorological variables such as 10m u-component of wind (u), 10m v-component of wind (v), surface pressure (PS), 2m air temperature ($T_{2m}$), and 2m dewpoint temperature ($TD_{2m}$), two moisture variables (soil volumetric water content of the 0–7 cm layer($\theta$),  and mean vertically integrated moisture

flux divergence (MFD)), and two cloud variables, including total cloud fraction and CBH, both of which are obtained from the hourly single level ERA5 reanalysis. In order to obtain the 2m relative humidity ($RH_{2m}$), the equation proposed by Lawrence (2005) is used, which is shown as follows:

$$RH_{2m} = 100 - 5 \times (T_{2m} - TD_{2m}) \,. \tag{1}$$

## 2.2 Methods

### 2.2.1 Pre-processing

Previous studies suggested that pure ice exists in part of the cloud when T drops below –20 °C (Minnis et al., 2005). Since radiosonde measurements only provide the RH profile with respect to liquid ($RH_{liquid}(z)$), the portion of radiosonde observed $RH_{liquid}(z)$ should be converted to RH profile with respect to ice ($RH_{ice}(z)$) for those observations at altitudes with T below –20 °C (Austin et al., 2009). According to Murray (1967) and Monteith and Unsworth (2008), $RH_{ice}(z)$ is converted from $RH_{liquid}(z)$ when T < –20 °C based on the following equations:

$$e_{liquid}(z) = 6.1078 \exp\left[\frac{17.2693882 T(z)}{T(z) + 237.3}\right], \tag{2}$$

$$e_{ice}(z) = 6.1078 \exp\left[\frac{21.8745584(T(z) - 3)}{T(z) + 265.5}\right], \tag{3}$$

$$RH_{ice}(z) = RH_{liquid}(z) \times \frac{e_{liquid}(z)}{e_{ice}(z)}, \tag{4}$$

where z represents altitude in units of km, T(z) represents the profile of air temperature in units of °C, $e_{liquid}(z)$ (hPa) and $e_{ice}(z)$ (hPa) denote saturation vapor pressure in the pure liquid and ice phase with respect to altitude, respectively. Note that the altitude represents the height AGL.

Note that besides equation (3), there are also several formulations for $e_{ice}$ (Murphy and Koop, 2005). In order to quantify the difference in $e_{ice}$, we also calculate the $e_{ice}$ for –40 to 0 °C using several equations listed in Murphy and Koop (2005), which are by Goff and Gratch (1946), Hyland and Wexler (1983), Sonntag (1990), and Marti and Mauersberger (1993). Obviously that the $e_{ice}$ calculated using different formulations are nearly the same (Fig. S1). Specifically, $e_{ice}$ calculated by Murray (1967) is mostly closed to that by Goff and Gratch (1946), with the absolute difference less than 0.004 hPa, followed by Hyland and Wexler (1983) and Sonntag (1990), with the absolute difference less than 0.009 hPa. The largest differences in $e_{ice}$ exist between Murray (1967) and Marti and Mauersberger (1993), reaching up to 0.012 hPa. These results could prove that our choice for $e_{ice}$ calculation is expected to affect the CVS results slightly.

When T is between –20 °C and 0 °C, liquid water and ice coexist in cloud (Austin et al., 2009). In this case, the RH is termed as $RH_{mixed}$, which is a combination of $RH_{liquid}$ and $RH_{ice}$. To obtain the $RH_{mixed}$ profile, the solutions of ice phase

and liquid phase are scaled linearly with $T(z)$ (Austin et al., 2009). Therefore, $RH_{mixed}$ at any given height z can be calculated by using the following equations:

$$RH_{mixed}(z) = \frac{-20 - T(z)}{-20} RH_{liquid}(z) + \frac{T(z)}{-20} RH_{ice}(z). \qquad (5)$$

In our retrieval method, the final RH profile $RH(z)$ used to derive the CVS can be described as $RH_{liquid}(z)$, $RH_{ice}(z)$, and $RH_{mixed}(z)$, depending on the phase state, which are calculated using the following formula:

$$RH(z) = \begin{cases} RH_{liquid}(z), & T \geqslant 0\ ℃ \\ RH_{mixed}(z), & -20\ ℃ < T < 0\ ℃. \\ RH_{ice}(z), & T \leqslant -20\ ℃ \end{cases} \qquad (6)$$

Before determination of CVS, the profiles of $RH(z)$ and $T(z)$ after the above pre-processing are smoothed by the arithmetical averages of $RH(z)$ and $T(z)$ at the altitudes of $z_{i-1}$, $z$, and $z_{i+1}$ ($i \geq 2$), respectively.

### 2.2.2 Determination of CVS

The presence of clouds greatly affects the $RH(z)$ and $T(z)$, which changes sharply when a sounding balloon enters or leaves cloud layers (Pietrowicz and Schiermeir, 1978; Matveev, 1981; Lawson and Cooper, 1990). Thus, the changes in the vertical gradients of $RH(z)$ and $T(z)$ can be used to identify cloud boundaries. Chernykh and Eskridge (1996) presented a method to determine cloud boundaries by using the second derivatives of $RH(z)$ and $T(z)$ ($RH''(z)$ and $T''(z)$), respectively. However, only using $RH''(z)$ and $T''(z)$ would cause us to misidentify cloud layers, tending to detect more cloud layers than observations, especially for very thin cloud layers (Zhang et al., 2012; Costa-Surós et al., 2014). To reduce this possibility, RH threshold should be combined to determine cloud layers. Zhang et al (2010) proposed altitude-dependent thresholds of RH to detect cloud layers. However, their method tends to identify less cloud layers (Costa-Surós et al., 2014). This is due to the application of slightly high RH thresholds, resulting in a lower detection rate of clouds.

In order to improve the accuracy of cloud detection, we develop a CVS retrieval method by combining the vertical gradients of $RH(z)$ and $T(z)$ and altitude-dependent thresholds of RH as well. After pre-processing the $RH(z)$, the first step in our method is to detect the bases and tops of moist layers. Generally, $RH(z)$ increases when a radiosonde balloon enters a moist layer, which suggests that the first derivative of $RH(z)$ is greater than zero ($RH'(z) > 0$). At the base of a moist layer, there is a jump in $RH(z)$ (Wang and Rossow, 1995), thus considering that $RH(z)$ reaches a local maximum increase ($RH''(z)$ < 0). At the same time, $T(z)$ stops decreasing (i.e., $T'(z) < 0$) near the base of a moist layer, due to the condensation of water vapor and its accompanying release of latent heat. Hence, $T(z)$ reaches a local minimum decrease, which means $T''(z) > 0$. Therefore, by examining the first and second derivatives of $RH(z)$ and $T(z)$ starting from the surface upward, the bases of moist layers can be detected when the following criteria are satisfied:

$$\begin{cases} RH'(z) > 0 \text{ and } RH''(z) < 0 \\ T'(z) < 0 \quad \text{and} \quad T''(z) > 0 \end{cases}$$ (7)

where $RH'(z)$ and $RH''(z)$ denote the first and second derivatives of $RH(z)$, respectively, and $T'(z)$ and $T''(z)$ denote the first and second derivatives of $T(z)$, respectively.

Similarly, the tops of moist layers are detected as follows:

$$\begin{cases} RH'(z) < 0 \text{ and } RH''(z) < 0 \\ T'(z) > 0 \quad \text{and} \quad T''(z) > 0 \end{cases}$$ (8)

To obtain $RH''(z)$ and $T''(z)$, Chernykh and Eskridge (1996) approximated radiosonde observed $RH(z)$ and $T(z)$ by cubic splines (Bartels et al., 1987), which leaded to bias in cloud detection. To avoid bias due to this approximation, we calculate $RH''(z)$ and $T''(z)$ as follows:

$$\begin{cases} RH'(z) = \dfrac{dRH(z)}{dz} = \dfrac{RH(z_{i+1}) - RH(z_i)}{z_{i+1} - z_i} \\ RH''(z) = \dfrac{d}{dz}\left(\dfrac{dRH}{dz}\right) = \dfrac{RH'(z_{i+1}) - RH'(z_i)}{z_{i+1} - z_i} \\ T'(z) = \dfrac{dT}{dz} = \dfrac{T(z_{i+1}) - T(z_i)}{z_{i+1} - z_i} \\ T''(z) = \dfrac{d}{dz}\left(\dfrac{dT}{dz}\right) = \dfrac{T'(z_{i+1}) - T'(z_i)}{z_{i+1} - z_i} \end{cases}$$ (9)

Then, we identify cloud layers from moist layers determined above using height-resolving RH thresholds defined in Table 2, which are lower than the thresholds proposed by Zhang et al. (2010) to reduce the restriction for a moist layer being identified as a cloud layer. The moist layer is identified as a cloud layer if the following three conditions are met: (1) the base of moist layer is greater than 280 m (Zhang et al., 2010); (2) the thickness of moist layer is larger than 30.5 m and 61 m for the base of moist layer less than 2 km and larger than 2 km, respectively (Poore et al., 1995; Zhang et al., 2010); (3) the minimum RH (min-RH) within the moist layer is greater than the corresponding min-RH threshold at the base of moist layer (Table 2); (4) the maximum RH (max-RH) within the moist layer is greater than the corresponding max-RH threshold at the base of moist layer (Table 2). By using max-RH, it is possible to avoid misidentifying some thin moist layer as cloud layer. Otherwise, the moist layer is discarded from the analysis.

**2.2.3 Post processing**

To obtain robust cloud structures, the cloud layers determined above have to be further reprocessed. If the distance between two contiguous cloud layers is less than 300 m, or the min-RH between the continues cloud layers is greater than the corresponding minimum RH threshold (inter-RH) between the consecutive cloud layers (Table 2), these two cloud layers are merged (Zhang et al., 2010).

The whole process, including the pre-processing of the RH profile, the determination of the cloud layers in vertical direction and the post-processing of detected cloud layers are schematically summarized in Fig. 2.

## 3. Results and discussion

### 3.1 Intercomparison of CVS from radiosonde, MMCR and ERA5

Before presenting the characteristics of near-global vertical structure of clouds, we first compare the CVS determined from radiosonde measurements by using our proposed retrieval method with those from the MMCR in Beijing. Figure 3 presents the comparisons of the cloud base and top heights of each layer at four selected cases. As seen from Fig. 3a, the one-layer cloud derived from radiosonde (left) agrees well with that obtained by MMCR (right), with the CBH (CTH) being 0.28 (9.57) km and 0.48 (9.66) km for radiosonde and MMCR, respectively. For high cloud case, the radiosonde-derived CBH and CTH are also consistent with those from MMCR (Fig. 3b). The radiosonde classifications of two-layer clouds are also in good agreement with those from MMCR (Figs. 3c and d). Above results indicate that the radiosonde has the potential to obtain accurate CVS for both low cloud and high cloud, assuming that there is no obvious change in CVS during short time (~15 min).

For the statistical analysis during the whole year of 2019 in Beijing (Fig. 4), the cloud base and top heights obtained by radiosonde are generally consistent with those from MMCR, with correlation coefficients (R) being up to 0.91 and 0.81, respectively. Note that to collocate radiosonde-derived CBHs (CTHs) with appropriate MMCR measured CBHs (CTHs), the time lag correction proposed by Dzambo and Turner (2016) is used. For low-level clouds (CBH < 2 km), the radiosonde-derived CBHs are obviously higher than those from MMCR. A possible reason is that MMCR detects cloud particles using millimeter-wavelength and is readily affected by the presence of large precipitation particles, as well as insects and bits of vegetation, which commonly suspended in the atmospheric boundary layer, resulting in lower cloud base (Zhang et al., 2013). For CTHs, the values retrieved from radiosonde are systematically higher than those from MMCR by about 0.86 km (Fig. 4b). There are three possible reasons for this difference. One is related to the variations in clouds from radiosonde observations caused by the horizontal drift of radiosonde balloons (Clothiaux et al., 2000; Zhang et al., 2013), which may explain the large discrepancies. Another reason is that MMCR is not sensitive to small cloud particles far from the radar, thereby tends to underestimate CTH (Zhang et al., 2019). The last one is that radiosonde tends to detect higher cloud top heights than those retrieved from MMCR due to delay (time lag) after the radiosonde balloon passes through a cloud layer because of the wetness of the sensors (Zhang et al., 2013). Overall, the results of Figs. 3 and 4 indicate that our proposed CVS retrieval method is reliable and can detect CVS accurately and reasonably.

Next, we compare the radiosonde-derived CVS over 2018–2019 with those from ERA5 reanalysis on a global scale, as shown in Fig. 5. In general, the CBHs from ERA5 tend to be lower than those from radiosonde, and their R values are 0.24 and 0.49 at 0000 UTC (Fig. 5a) and 1200 UTC (Fig. 5b) for the whole data, respectively. This result is similar with that reported by Li et al (2022), which demonstrated that the differences in CBHs between ERA5 and the surface observation are

much higher at 0000 UTC. The reason for that the correlation coefficient at 1200 UTC is more than twice as large as at 0000 UTC is complicated, which may be associated with the uncertainties of RH and T profiles, and the assimilation windows (within 12 h) for model constraint when producing hourly ERA5 data (Hersbach et al., 2020). At a more detailed level, the CBHs from ERA5 tend to be much lower than those from radiosonde for clouds with CBHs from radiosonde > 6 km. Previous studies reported that the underestimation for high-level clouds in the ERA5 results were caused by the poor specification or parameterization of critical RH from model (Miao et al., 2019). To clearly examine the case where most data exist in Figs. 5a and b, the comparison for CBHs from radiosonde < 2 km is further given at Figs. 5c and d. Compared to the results of all the cloud layers, the fitting slopes of the low cloud layers are slightly smaller and the R values are smaller by about 25%. This indicates that the CBHs from ERA5 are roughly underestimated compared with radiosonde observations at relatively low altitude. The reason may be associated with issues of cloud parameterizations schemes used for ERA5. The CBH in ERA5 is detected using the cloud cover or cloud water mixing ratio threshold. When cloud cover is greater than 1 %, the height from ground is defined as CBH (Wang et al., 2022), which may lead to the underestimation of CBH in ERA5. Note that there exist several cases with very low CBH from ERA5 (e.g., CBH < 0.1 km) which is not reasonable.

## 3.2 Statistic analysis of cloud occurrence frequency

The annual and seasonal frequency distributions of clouds with the number of layers at 1200 UTC over 2018–2019 are presented in Fig. 6. On the annual time scale, a total of 201,558 profiles from radiosonde are retrieved. Of all detected profiles, 30.5 % cases are clear skies, indicating 69.5 % of the time is cloudy (Fig. 6a), which is in good agreement with the global mean cloud fraction (about 68.0 % ± 3.0 %) reported by Stubenrauch et al. (2013), who analysed the global cloud fraction using multiple satellite observations (e.g., MODIS, MISR, POLDER and CALIPSO). Over cloudy skies, the cloud frequency decreases as the number of layers increases (Fig. 6a), with the one-, two-, three-, four-, five-, six-, and seven-layer clouds accounting for 53.8 %, 28.0 %, 11.4 %, 4.2 %, 1.4 %, 0.6 %, and 0.4 %, respectively. Wang et al. (2000) and Subrahmanyam and Kumar (2017) found similar decreasing trends by using global radiosonde measurements (vertical resolution of ~100 m) and global satellite observations (CALIOP), and they demonstrated that 58 % of clouds were one-layer clouds. However, the occurrence frequencies of multi-layer clouds in our study are obviously higher than those provided by Subrahmanyam and Kumar (2017), which reported that two-, three-, four-, and five-layer clouds occurred at frequencies of 20.00 %, 3.50 %, 0.40 %, and 0.04 %, respectively. The possible cause for this difference could be that satellite has insufficient detection capabilities for multi-layer clouds, leading to an underestimation of their occurrence frequencies (Chang and Li, 2005).

Among the multi-layer clouds, 61.1 % are two-layers clouds. This result is slightly lower than that reported by Wang et al. (2000), who suggested that about 67 % of the multi-layer clouds were two-layer clouds. The possible reasons are the discrepancies in the study periods, the spatial distributions of radiosonde stations, the vertical resolution of the radiosonde measurements, and the CVS derivation methods. At 0000 UTC, cloudy skies occur 65.3 %. Of all cloud figuration, the occurrence frequency of one-layer clouds accounts for 55.4 %. Among multi-layer clouds detected, 62.2% are two-layer clouds (Fig. S2a).

The seasonal frequency distributions of clouds with the number of layers (Figs. 6b, c, d, and e) are similar with those at annual time scale, namely, frequency decreases as the number of layers increases. By comparison, more clouds occur in December–January–February (DJF; 72 %) and March–April–May (MAM; 70.8 %) than do in June–July–August (JJA; 69.3 %) and September–October–November (SON; 66.3 %). These results are consistent with those from Xi et al. (2010) based on a 10–year climatology of cloud fraction from surface observations. The seasonal variations are caused by the fact that cloud fractions vary monthly, with a maximum of 83 % in February and a minimum of 65 % in September (Fig. S3b). For one-layer clouds, the occurrence frequency is highest in MAM and DJF and lowest in JJA. The possible reason is that the radiosonde sites in this study are mainly located in northern hemisphere (NH) (Fig. 1) and more stratus clouds occur in boreal winter (DJF) and boreal spring (MAM) than boreal summer (JJA) (Dong et al., 2005). In contrast, the occurrence frequency of multi-layer clouds is highest in boreal summer (JJA), owing to the deeper troposphere and more convective storms in summer (Zhang et al., 2010). The frequencies of clouds with various layers at 0000 UTC on seasonal scale (Figs. S2b, c, d, and e) are also similar with those at 1200 UTC to some degree.

### 3.3 Near-global vertical distribution of cloud

The annual mean CVS for one-, two-, three-, four- and five-layer clouds at 1200 UTC are shown in Fig. 7a. In general, the CBH (CTH) decreases slightly (increases significantly) as the number of cloud layers increases. The CBHs of one-, two-, three-, four-, and five-layer cloud are $3.58 \pm 3.31$, $2.61 \pm 2.76$, $2.07 \pm 2.44$, $1.92 \pm 2.49$, and $1.97 \pm 2.71$ km, and the corresponding CTHs are $6.52 \pm 3.94$, $9.42 \pm 3.04$, $10.71 \pm 2.94$, $11.81 \pm 3.04$, and $13.17 \pm 3.06$ km, respectively. This result is consistent with Zhang et al. (2010), who reported that one-layer clouds are roughly located at altitudes that fall somewhere among the altitudes of the multi-layer cloud configurations. It is obvious that the lowermost cloud layer in multi-layer clouds occurs below 3 km with slight variations, while the upper layer exhibits more significant variations. Similarly, Wang et al. (2000) pointed out that the lowermost layers of two- and three-layer clouds occur mostly below 3 km, and the uppermost layer of two- and three-layer clouds occurs over a wide range of heights centered in 6 to 7 km, and 7 to 8 km, respectively. Interestingly, the thickness of one-layer clouds ($2.94 \pm 3.00$ km) is greater than that of multi-layer clouds of any number of layers ($0.67 \pm 0.88 \sim 2.44 \pm 2.07$ km), and the thickness of each cloud layer for multi-layer clouds decreases as the number of cloud layers increases, which is consistent with previous studies (e.g., Luo et al., 2009; Chi et al., 2022). For multi-layer clouds, the thickness of the lowermost layer is larger than that of the upper layer. These CVS results are mainly attributed to the combined effects of solar heating and exchange of longwave radiation between surface and cloud layers (e.g., Rogers and Koracin, 1992; Guan et al., 1997; Zhang et al., 2010). Note that for the three-, four-, and five-layer cloud configurations the maximum distance between two continues cloud layers is around 2.5 km, existing between the first and second lowest layer. For the upper layers, the distance between two continues cloud layers decreases to be about 1.0 km.

To further quantify the CVS for one- and multi-layer clouds, their boxplots are shown in Fig. 7b. Overall, for multi-layer clouds, the mean CBH is lower than that for the one-layer clouds by 1.20 km, while the mean CTH (CT) is much greater than that for one-layer by 3.47 km (4.67 km). The possible reasons are that multi-layer clouds often occur at humid climate regions

(e.g., southeast of China and east of USA as shown in Figs. 12c, d, e, and –f) and the air parcel tends to reach the lift condensation level (LCL) at relatively low altitude, resulting in lower CBH compared to one-layer clouds. Most multi-layer clouds are generated from one-layer clouds by extending the range of CBH and CTH of one-layer clouds. Thus, we expect that the CTH and CT of multi-layer clouds are larger than that of one-layer clouds. Note that the different layer cloud configurations occur at different frequencies as shown in Fig. 5a.

The CTH and CBH are two of the most important CVS parameters that play a significant role on estimating CRE at the top of atmosphere and the surface, respectively (Wang and Rossow, 1995; Loeb et al., 2012; Xu et al., 2021b). In this study, we present the vertical distributions of annual and seasonal occurrence frequencies of CBH, CTH, and cloud at 1200 UTC in Fig. 8. As references for altitude, the annual and seasonal mean planetary boundary layer height (PBLH) and tropopause height retrieved from radiosonde observations are also given in Fig. 8. Generally, these results of the vertical occurrence frequencies of clouds at annual scale are close to those at seasonal scale. It is obvious that the relatively large occurrence frequencies of CBH occur within 1 km, with the highest frequency being at about 0.5 km (Figs. 8a and d), which is lower than the annual mean PBLH (~0.76 km). This indicates that most of cloud bases exist in the atmospheric boundary layer as previous studies (Zhang et al., 2014). Above the top of the boundary layer, the occurrence frequencies of CBH decrease with altitude, since most clouds are suppressed by the inverse layer at the top of boundary layer (Sugimoto et al., 2000). Over 6 km, the occurrence frequency of CBH in boreal summer (JJA) is higher than that in other seasons (Fig. 8d). The reason may partly be that more cirrus clouds occur in summer (Wang et al., 2000). Below 1.5 km, the occurrence frequency of CBH is the highest during in boreal winter (DJF). This phenomenon can be explained by the combined effects of low temperature and upward-motion conditions in winter, which promotes condensation or collision/coalescence of water vapor to form cloud droplets at relatively low altitudes during the ascending motion, resulting in low cloud bases (Dong et al., 2005; Chi et al., 2022). These results in the occurrence of CBH generally peak in boreal summer (JJA) and boreal spring (MAM) and reach its lowest value in boreal winter (DJF) and boreal autumn (SON) are in good agreement with previous studies (e.g., Dong et al., 2005; Xi et al., 2010; Zhang et al., 2019).

Different from CBH, the occurrence frequencies of CTH exhibit a bimodal distribution, with a lower peak between 0.5 and 3.0 km and an upper peak between 6 and 12 km (Figs. 8b and e), which agrees with previous studies based on ground-based lidar and radar measurements (Comstock and Jakob, 2004). The occurrence frequency of CTH reaches a maximum at around 11 km, which is slightly lower than the corresponding global annual mean tropopause height (almost 12 km); and then it decreases rapidly with altitude. Above 9 km, the maximum occurrence frequency of CTH occurs in boreal summer (JJA), which is due to the deeper troposphere and the higher frequency of deep convective clouds in summer (Johnson et al., 1999; Zhang et al., 2019). This result is consistent with that reported by Zhang et al. (2019), which demonstrated that CTH peaked around in summer. Below 3 km, the occurrence frequency of CTH is the highest in boreal winter (DJF). The occurrence frequencies of CTH tend to increase with altitude between 3 and 9 km irrespective of seasons.

As for CTH, the vertical distribution of cloud occurrence frequency is distinctly bimodal, but in different ranges of altitude, namely, a lower peak between 0.5 and 3 km and a higher peak between 6 and 10 km (Figs. 8c and f). These findings are in line

with previous studies (e.g., Mace and Benson, 2008; Zhang et al. 2014), which reported that the vertical profile of cloud occurrence peaked both in the boundary layer and upper troposphere by analysing cloud retrievals from radiosonde, models, and satellites at both regional and global scale. Above 6 km, more clouds occur in boreal summer (JJA) than in other seasons, while below 3 km, there are more clouds in boreal winter (DJF).

To further study the characteristics of CVS over different regions of the world, the vertical distributions of the occurrence frequencies of CBH, CTH, and cloud over six regions of interest (ROI; Fig. 1) at 1200 UTC are shown in Figs. 9, 10, and 11. The regional annual mean PBLH and tropopause height as functions of longitude and latitude are also shown. Similar with the results of global mean CBHs (Figs. 8a and d), the maximum occurrence frequencies of regional mean CBHs are also within the regional mean PBLH (Figs. 9a, b, c, g, h, and –i). The occurrence frequencies of CBHs are nearly zero above the tropopause. The conditions that cloud bases above the tropopause are often results of the overshooting tops from strong convective storms, such as deeply penetrating cumulonimbus and thunderstorms (Rosenfeld et al., 2007; Homeyer and Kumjian, 2015; Liu et al., 2021). Due to strong upward motion contained in the strong convective storms, the overshooting tops can reach as high as 19– 20 km (Hassim et al., 2014). The frequency of cloud bases between 1 and 3 km in East Asia is roughly larger than that over North America and Europe, which is consistent with previous studies (e.g., Zhang et al., 2013; Zhou et al., 2021; Sharma et al., 2022). The possible reason is that the air pollution issue in China is more severe, and the higher aerosol loading can change the macro- and micro-physical properties of clouds and may invigorate convective clouds at high altitudes at the expense of low-level clouds (Wall et al., 2014; Guo et al., 2018; Liu et al., 2020). The variation pattern of CBH frequency with altitude in the Southern Hemisphere (SH) is similar with that in NH, but its fluctuation is roughly greater due to fewer stations available there. A close inspection of the horizontal distribution of CBHs shows that there is a west-high-east-low pattern in America and a north-high-south-low pattern in China (Fig. 9e), which correspond to arid (high CBHs) and humid (low CBHs) climates there. Most (67 %) annual mean CBHs range from 2 to 4 km with an average of $3.07 \pm 1.06$ km (Figs. 9d, e, and f).

The characteristic of CTHs over the six ROIs at 1200 UTC are shown in Fig. 10. Similar with the global mean CTHs, the regional mean occurrence frequencies of CTHs above tropopause also decrease rapidly with altitude. As expected, most of cloud tops are within the tropopause height in East Asia and North America. However, this predominance is not obvious in Europe and Antarctica, where the tropopause is lower since the climates are humid. In East Asia, the CTH frequency is generally highest in boreal summer (JJA) above 6 km since there is more frequent deep convective cloud during the summer monsoon (Wang et al., 2000), while highest in boreal winter (DJF) below 6 km as few solar radiations reach the surface in winter that the cloud development is weak, thus the CTH is relatively low. This result is in agreement with previous cloud radar observations (Zhang et al., 2019). Relatively large CTHs occur at southern USA and southeast China which are affected by monsoon, resulting in more deep convective clouds (Fig. 10e). Note the mean CTHs at tropic western Pacific are significantly larger than other regions, reaching up to 12 km. At this region, the occurrence frequency of subvisible cirrus is highest (data not shown). This can be explained that deep convection mostly occurs at tropic areas, favoring the formation of subvisible cirrus (Krämer et al., 2009; Froyd et al., 2010). These results are consistent with previous studies based on satellite

observations (Martins et al., 2011; Schoeberl et al., 2022). Most (85 %) annual mean CTHs vary from 7 to 12 km with an average of 8.44 ± 1.52 km (Figs. 10d, e, and f).

The annual and seasonal mean occurrence frequencies of clouds at different regions are shown in Fig. 11. Bimodal distribution of the occurrence frequencies of clouds with altitude appears at North America and Europe, peaking at the top of boundary layer (~1 km) and the tropopause (10~11 km). While unimodal distribution is at East Asia, peaking at about 3 km, namely fewer clouds occur in the boundary layer compared to the result of global mean. This is attributed to more serious air pollution issue in East Asia compared to North America and Europe, resulting in higher occurrence frequencies of convective clouds at high altitudes (Liu et al., 2020; Sharma et al., 2022). This result is consistent with Xu et al. (2021a). Above 6 km in East Asia (~9 km at North America; ~10 km at Europe), the cloud occurrence frequency in JJA is higher than those in other seasons, because the deeper troposphere and the more frequent occurrence of convective storms in summer (Xi et al., 2010). Below 3 km in East Asia (~1 km in North America and Europe), the frequency of cloud occurrence in DJF is highest, since stratus clouds usually occur in winter (Dong et al., 2005). In Antarctica, the peak of cloud occurs at low altitude (~1 km, a little bit higher than the PBLH ~ 0.70 km) due to the cold climate there. Contrary to CBHs, the annual mean CTs have a west-lower-east-higher pattern in North America, and a northwest-lower-southeast-higher pattern in East Asia. Relatively large CTs exist in Europe since there are sufficient water vapours for cloud development under the oceanic climate. The CTs at 20–30 ° zone are relatively smaller (Fig. 11f), because there coincide with the mean subsidence zone of the Hadley circulation (Poore et al., 1995). Most (52 %) annual mean CTs vary from 4 to 6 km with an average of 5.37 ± 1.58 km (Figs. 11d, e, and f).

The corresponding near-global and regional mean vertical distributions of occurrence frequencies of CVS at 0000 UTC on annual and seasonal time scales (Figs. S4, 5, 6, 7, and 8) are similar with those at 1200 UTC.

## 3.4 Near-global horizontal distribution of cloud

Figure 12 presents the horizontal distribution of the annual mean occurrence frequencies for different layer clouds. Generally, more clouds occur at humid climate regions (e.g., eastern USA, southern China, and western Europe) (Fig. 12a). This is due to sufficient water vapor supply from the surface, and clouds can be generated if aerodynamic and thermodynamic conditions required for cloud decoupling are met. Similarly, the wetter the climate, the higher the occurrence frequencies of clouds for three-, four- and five-layer clouds in cloudy skies (Figs. 12d, e, and f). On the contrary, more one-layer clouds in cloudy skies occur at regions of relatively arid climate compared to the humid climate regions (Fig. 12b). Reddy et al. (2018) reported the occurrence of multi-layer clouds was more significant under moist atmospheric conditions, while more one-layer clouds occurred under dry atmospheric conditions. As the transition between one- and multi-layer clouds, the spatial feature of two-layer clouds (Fig. 12c) is not apparent compared to other cloudy conditions. Spatially, more one-layer clouds occur over Asia than in North America (Fig. 12b), while there are roughly larger cloud frequencies for multi-layer clouds over North America (Figs. 12c, d, and f). These results are consistent with Wang et al. (2000), which demonstrated that the frequency of multi-layer clouds was higher in North America than in Asia. Few studies provided the global spatial distribution of the occurrence

frequencies of clouds with various number of layers (from one- to five-layer) by radiosonde measurements as shown in Fig. 12.

Figure 13 shows the horizontal distributions of the near-global CBH, CTH, and CT at 1200 UTC. In terms of seasonality, the CBHs in boreal winter (DJF) vary little with space compared to other seasons. The magnitudes of CBH follow the order of boreal summer (JJA: 3.74 ± 1.56 km) > autumn (SON: 3.08 ± 1.22 km) ≈ spring (MAM: 3.05 ± 1.11 km) > winter (DJF: 2.55 ± 1.02 km) (Figs. 13a, b, c, and d). This result is consistent with previous satellite observations that the CBHs reached its maximum in summer and minimum in winter (Chi et al., 2022). The reason can be explained that more solar radiation energy is available for cloud development in summer (Zhang et al., 2018). In East Asia, the CBHs exhibit a northwest-high-southeast-low pattern, namely, high clouds (as high as 4.0 km) tend to occur in northwest China while low clouds (less than 2.0 km) tend to occur in southeast China and the northern part of the South China Sea (Figs. 13a, b, c, and d). This phenomenon further illustrates that humid (arid) climate tends to generate low (high) clouds. The reason may be due to that the higher dew point at humid climate regions always cause a lower LCL, resulting in lower cloud bases (Zhang et al., 2018). Similarly, in north America, the CBHs are lower in the east and higher in the west, which is consistent with the results of An et al. (2017) based on the information from the Automated Surface Observing System Observation for a 5–year period (2008–2012). This can be explained that the western USA is often dry near the surface, thus PBLHs tend to be much deeper, resulting in higher CBHs compared to eastern USA.

For CTH, its relative magnitudes have the same seasonality with those for CBH, namely, higher CBH occurs in boreal summer (JJA:9.24 ± 1.67 km) and relatively lower value occurs in boreal winter (DJF: 7.63 ± 2.05 km) (Figs. 13e, f, g, and h). This is because more deep convective clouds occur in summer compared to other reasons (Johnson et al., 1999; Zhang et al., 2019). During boreal summer (JJA), the spatial distribution of CTHs exhibit an obvious southeast-high-northwest-low pattern in China, and an east-high-west-low pattern in contiguous United State. The phenomenon of the southeast-high-northwest-low pattern in China is much related to the East Asia summer monsoon, which causes strong atmospheric convection in the southeastern China and makes the convective instability layer much thicker. This allows the water vapor to be transported to a higher altitude and favors the production of deep convective clouds (Sun et al., 2019; Chi et al., 2022). The east-high-west-low pattern of CTH in contiguous United State is attributed to the humid (arid) climate in the east (west) of United States.

The CT values (Figs. 13i, j, k, and l) undergo almost the same seasonality with the CBHs and CTHs with the order of JJA (5.50 ± 1.89 km) > SON (5.23 ± 1.58 km) ≈ MAM (5.20 ± 1.59 km) > DJF (5.08 ± 1.96 km). The spatial pattern of CT is similar with that of CTH with larger CT at the humid climate regions and smaller CT at the arid climate regions. The reason is that higher RH not only contributes to a lower cloud base but also helps an air parcel to reach higher levels, and thus leads to larger cloud thickness (Zhang and Klein, 2013). Interestingly, the CTs in Europe are relatively large compared to those in China and USA. As mentioned before, this is partly explained by that oceanic climates dominate in Europe where more water vapor supply is available for cloud development.

Overall, the spatial distributions of CVS characteristics in different seasons are largely affected by the climates. The arid (humid) climates favor larger (smaller) CBH, smaller (larger) CTH and CT. The corresponding results of CVS at 0000 UTC are roughly close to those at 1200 UTC, as shown in Figs. S9 and 10.

**3.5 Diurnal variation of cloud occurrence frequency at various heights**

Since near-global radiosonde sites cover almost all time zones, the diurnal cycle of cloud vertical structures can be obtained
by converting UTC to local solar time (LST). Based on the radiosonde-derived CVS, the diurnal variations of height-resolved occurrence frequencies of clouds in boreal summer (JJA) and boreal winter (DJF) are presented in Fig.14. The mean vertical profiles of the occurrence frequency of liquid, ice, and mixed clouds at both 0000 and 1200 UTC are also shown for the corresponding season. Ice clouds are identified when T below -20 °C, liquid clouds are detected with T above 0 °C, and mixed clouds correspond to T between -20 °C and 0 °C, as described in the Section 2.2.1.

In boreal summer, the maximum occurrence frequencies of clouds mainly appear at altitudes of 0.5–3 km and the second peak extend upward to be 6–10 km (Fig. 14b). The lower peak of the cloud occurrence frequency (0.5–3 km) roughly corresponds to the peak of the liquid cloud occurrence frequency (0.5–3 km), while the upper peak of the cloud occurrence frequency (6–12 km) roughly corresponds to the combined peak of the ice (8–12 km) and mixed cloud occurrence frequencies (6–8 km) as shown in Figs. 14b and c and Figs. 8c and f. Note that the vertical distribution of liquid and ice clouds occurrence
frequencies are expected, since T tends to decrease as altitude increases and thus, ice clouds tend to be generated at higher altitude compared to liquid clouds. Previous studies also reported that ice clouds were located higher than liquid clouds based on satellite data (Chang and Li, 2005). Especially, in the tropical and midlatitudes, subvisible cirrus clouds can reach to the upper tropopause (Gierens and Spichtinger, 2000; Immler et al., 2008). Subvisible cirrus generally occurs at the ice supersaturated regions, and can form in situ or as a consequence of deep convection (Krämer et al., 2009; Froyd et al., 2010).
The clouds in the lower atmosphere tend to appear as sun rises and increases from morning to early afternoon (1500 LST) within 3 km altitude. The reason is that, during daytime solar radiation heats the surface and drives turbulence and convection in the boundary layer, which largely affects the boundary layer clouds (Noel et al., 2018). In later afternoon (1600–1800 LST), clouds tend to form most frequently at relatively high altitudes (6–12 km), which is consistent with Chen et al. (2018). The reason may be that after solar radiation reaches peak, the sun heats less and cloud top begins to cool, which increases the
atmospheric instability, and fuels the development of one-layer cloud and the uppermost layer of cloud (Zhang et al., 2010). These findings are consistent with those reported by Chang and Li (2005) who analysed satellite cloud retrievals on a global scale. After 1800 LST, clouds tend to occur below 3 km again.

        For boreal winter, clouds mainly occur with maximum frequencies at altitudes of 0.5–3 km, which roughly corresponds to the combined peak of the liquid (0.5–3 km) and mixed cloud occurrence frequencies (0.5–6 km) as shown in Figs. 14e and
f and Figs. 8c and f. Note that there is a relatively large occurrence frequency of clouds during 0600–0800 LST, which is consistent with Betts and Tawfik (2016), who demonstrated that clouds had a near-sunrise peak in the cold season. The reason

may be that the air temperatures are lowest around sunrise for a stable boundary layer, which are beneficial for water vapor to accumulate and form cloud (e.g., Dai, 2001; Eastman and Warren, 2014; Gao, et al., 2019; An et al., 2017; 2020).

In boreal spring and autumn, the characteristics of the diurnal variations of cloud occurrence frequencies falls between that in boreal summer and winter. The diurnal variation of height-resolved occurrence frequencies of clouds provided by this study can be a reference for satellite-retrieved CVS products, even though the sampling at each hour is not even (Figs. 14a and d).

### 3.6 Association of CBH with meteorological variables and moisture

We further investigate the potential key factors linked to CBH, since CBH is one of the most important factors affecting CRE in numerical weather forecast and regional climate model (Kato et al., 2011; Viúdez-Mora et al., 2015; Prein et al., 2015). Previous studies suggested that CBH is affected by thermodynamic conditions (e.g., surface T, RH, and PS) and large-scale dynamic process (wind speed) at the near surface layer, and moisture conditions (Mauger and Norris, 2010; Gbobaniyi et al., 2011). In this study, we investigate the relationship of CBH with meteorological variables ($T_{2m}$, PS, $RH_{2m}$, and $WS_{10m} = \sqrt{u^2 + v^2}$), and moisture ($\theta$ and MFD).

The spatial distributions of the correlation relationships between CBH and meteorological variables are displayed in Fig. 15. Each point is the R value of a given CBH and meteorological variable at 1200 UTC over 2018–2019. Also given are the mean value and one standard deviation of the Rs on a global scale, which is obtained by averaging the absolute R value of each station. Generally, the magnitudes of the global mean absolute R values follow the order of R (CBH & $RH_{2m}$) = 0.29 ± 0.17 > R (CBH & $T_{2m}$) = 0.22 ± 0.16 > R (CBH & PS) = 0.17 ± 0.14 > R (CBH & $WS_{10m}$) = 0.15 ± 0.13. Since the CBH is detected using the profiles of T and RH, it is reasonable to expect that the correlation between CBH and $T_{2m}$ and $RH_{2m}$ are relatively high. Our result is consistent with Gbobaniyi et al. (2011), which demonstrated that surface temperature and specific humidity were strongly coupled with CBH. Interestingly, the CBHs are negatively related with $RH_{2m}$ while positively related with $T_{2m}$, which further convince us that humid and cold climate tends to generate low clouds. The physical explanation could be that, compared to dry air, humid air tends to reach condensation at relatively low altitude, thus the corresponding CBH is low. Note that the correlation between $T_{2m}$ ($RH_{2m}$) and CBH tends to be larger in northern China compared to the other regions of China. This is attributed to the combined effects of the larger $T_{2m}$ and smaller $RH_{2m}$, resulting in larger CBH (as shown in Figs. 13a, b, c, and d). Compared with $T_{2m}$ and $RH_{2m}$, the correlation relationship between CBH and PS and $WS_{10m}$ are relatively weaker (most of absolute R < 0.3). Note that the correlation of CBH and $WS_{10m}$ is negative in southeastern China but positive in northwestern China. The reason may be that the climate of southeastern China is mainly affected by monsoon and the large WS brings more humid air, causing lower cloud bases; while the air is relatively dry in the northwestern China, a region that is largely affected by the high pressure from Siberia, and large WS brings more dry air, corresponding to higher cloud bases.

Figure 16 shows the spatial distributions of the correlation relationships between CBH and soil moisture ($\theta$, left panels) and MFD (right panels). Obviously, most CBHs are negatively correlated to $\theta$ with mean absolute R (CBH & $\theta$) = 0.22 ± 0.15. This suggests that large soil moisture corresponds to low CBH, since more surface available energy are partitioning to latent heat flux with a large soil moisture, causing a reduced boundary layer height. Thus, clouds are generated with low bases (Betts, 2004; Cook et al., 2006; Huang and Margulis, 2013). The correlation of MFD with CBH is relatively weak, with most R values ranging from – 0.1 to 0.1 (Fig. 16b). This is not surprising since CBH is not significantly affected by the transportation of water vapor. Overall, the correlation between CBH and $RH_{2m}$ is the strongest, followed by $T_{2m}$, $\theta$, PS, and $WS_{10m}$, and MFD is the least.

To further obtain a quantitative understanding of the effects of meteorological variables and moisture on CBH, the 2D joint distribution of CBH is presented in Fig. 17. At given $T_{2m}$, larger $RH_{2m}$ corresponds to lower CBH; while at given $RH_{2m}$, larger $T_{2m}$ tends to result in higher CBH (Fig.17a). Besides, the combined effect of $T_{2m}$ and $RH_{2m}$ on the variation of CBH shows that larger $T_{2m}$ and smaller $RH_{2m}$ result in higher CBH. The same as Fig. 16, the CBH varies slightly as MFD changes, indicating that MFD exerts little impact on CBH. Different with MFD, larger $\theta$ attributes to lower CBH when $\theta > 0.2$ m$^3$ m$^{-3}$. However, CBH is not sensitive to low $\theta$. The relationship of CBH with meteorological variables and soil moisture at 0000 UTC (Figs. S11, 12, and 13) are similar to those at 1200 UTC.

## 4. Summary and conclusions

Based on high-vertical-resolution radiosonde observations from 374 near-global radiosonde stations, two years' worth (2018–2019) of near-global, high-quality CVS has been determined. A novel retrieval method is developed which combines the vertical gradients of T and RH and the altitude-resolved thresholds of RH. The accuracy of radiosonde derived CVS is assessed by comparison with MMCR measurements at Beijing site during the year of 2019. The good agreement in CBHs (R = 0.91) and CTHs (R = 0.81) confirms that this retrieval method performs reasonably well. On global scale, the CBHs from ERA5 tend to be lower than those from radiosonde with R of 0.24 and 0.49 at 0000 and 1200 UTC, respectively. The characteristics of near-global CVS are summarized as follows:

The near-global annual mean occurrence frequency of all clouds is 65.3 % (69.5 %) at 0000 (1200) UTC, which is close to the result from multiple satellite observations (68.0 % ± 3.0 %). Over cloudy skies, the cloud frequency decreases as the number of cloud layers increases. The one-layer clouds are predominant accounting for 55.4 % (53.8 %) of all cloud configurations. Among the detected multi-layer clouds, 62.2% (61.1%) are two-layer clouds at 0000 (1200) UTC. More clouds occur in boreal winter (DJF) and spring (MAM) than in summer (JJA) and autumn (SON).

The CBH and CT of each cloud layer (CTH) tend to decrease (increase) as the number of cloud layers increases. For one-layer clouds, they are roughly located at altitudes that fall somewhere among the altitudes of the multi-layer cloud configurations, and their thicknesses are greater than those of any layer of multi-layer clouds. For multi-layer clouds, the lowermost cloud layer almost occurs below 3 km with small variation, while the upper layer exhibits more significant variation.

The maximum altitude difference between cloud layers is around 2.5 km, which exists between the first and second layer for the three- and more multi-layer cloud configurations. For the upper layers, altitude difference decreases to be about 1.0 km.

In most of cases CBH are located within 1 km with the highest frequency at 0.5 km, which is lower than the global annual mean PBLH (0.76 km). This result indicates that in most of the time cloud bases are in the atmospheric boundary layer. The vertical distributions of the CTH occurrence frequencies have two peaks with the upper peak at around 11 km, slightly lowering

than the global annual mean tropopause height (11.76 km). Similar with CTHs, the cloud occurrence frequencies also exhibit a bimodal distribution, with a lower peak being between 0.5 and 3 km and an upper peak being between 6 and 10 km, agreeing well with previous studies based on radiosonde, models, and satellites at both regional and global scale. Regionally, the occurrence frequencies of CBH in East Asia between 1 and 3 km is generally larger than those over North America and Europe, probably due to the severe air pollution issue in East Asia that invigorates convective clouds at high altitudes which suppresses

the occurrence of low-level clouds. The mean CTHs are highest at tropic western Pacific, where subvisible cirrus mostly occurs.

As for the horizontal distribution of CVS, we find that there are more clouds at humid climate regions (e.g., eastern USA, southern China, and western Europe) since sufficient water vapor supply from the surface benefits the generation of clouds. More one-layer clouds occur at arid climate regions, while more clouds with more than two layers occur at humid climate regions. Compared to one-layer clouds and multi-layer clouds, there is no apparent spatial feature for two-layer clouds.

The global mean CBH, CTH, and CT are 3.15 ± 1.15 (3.07 ± 1.06), 8.04 ± 1.60 (8.44 ± 1.52), and 4.89 ± 1.36 (5.37 ± 1.58) km at 0000 (1200) UTC, respectively. The CBH, CTH, and CT have almost the same seasonality, with magnitudes following the order of boreal summer > autumn ≈ spring > boreal winter. The spatial distributions of CVS are largely affected by the climate, namely, the arid (humid) climates correspond to larger (smaller) CBH, smaller (larger) CTH and CT.

In terms of the diurnal variation of cloud occurrence frequency, in boreal summer the maximum cloud occurrence

frequencies mainly appear at the altitudes of 0.5–3 km, corresponding to the peak of the occurrence frequencies of liquid clouds; and the second peak extends upwards to be 6–12 km, corresponding to the combined peak of the occurrence frequencies of ice and mixed clouds. During daytime, clouds are affected by solar radiation, tending to increase from morning to early afternoon (1500 LST) within 3 km altitude, peak in the later afternoon (1600–1800 LST) at relatively high altitude (6–12 km), and then decrease after 1800 LST. In boreal winter, clouds are relatively low with maximum frequencies appear at altitudes of

0.5–3 km, corresponding to the combined peak of liquid and mixed cloud occurrence frequencies. And the clouds have a peak during the period 0600–0800 LST, associating with the lowest temperature around sunrise, which are beneficial for water vapor to accumulate and form clouds. The seasonal diurnal variations of cloud occurrence frequency are in accordance with those from satellite and ground measurements.

As for the correlation of CBH with meteorological variables and moisture, we find that the CBH is correlated with both

the $RH_{2m}$ and $T_{2m}$, with global mean absolute R values of 0.29 ± 0.17 and 0.22 ± 0.16, respectively. That is, larger air temperature and smaller humidity tend to result in larger CBH. Most CBH is negatively correlated to soil moisture with global mean absolute R value of 0.22 ± 0.15 indicating larger soil moisture corresponds to lower cloud bases. The relationship between CBH and moisture flux divergence is weak.

This study is the first attempt to characterize the near-global CVS from high-vertical-resolution radiosonde data. These results of CVS could be used to validate output from global climate models, as constraints for cloud formation would need to be done inside a parameterization. In addition, these results can also be readily used by the operational weather community given the widespread use of radiosondes in day-to-day forecasting operations. The formation mechanism of cloud is complex, studying features affecting the CVS appears to be a promising avenue for future work.

*Code/Data availability.* The authors would like to acknowledge the National Meteorological Information Centre (NMIC) of CMA, NOAA, German Deutscher Wetterdienst (Climate Data Center), UK Centre for Environmental Data Analysis (CEDA), GRUAN, and the University of Wyoming (http://data.cma.cn/en, CMA, 2023; https://www.ncei.noaa.gov/data/us-radiosonde-bufr/archive/, NOAA, 2023; https://opendata.dwd.de/climate_environment/CDC/ observations_germany/radiosondes/high_resolution/historical/, Deutscher Wetterdienst (Climate Data Center), 2023; https://catalogue.ceda.ac.uk/, CEDA, 2023; ftp://ftp.ncdc.noaa.gov/pub/data/gruan/processing/level2/RS92-GDP/version-002/, GRUAN, 2023; and http://weather.uwyo.edu, The University of Wyoming, 2023) for providing the high-vertical-resolution radiosonde data. We would also like to thank the CMA for Ka-band millimeter-wave cloud radar (MMCR) data (https://data.cma.cn/en, CMA, 2023), and the ECWMF for ERA5 data (https://cds.climate.copernicus.eu/cdsapp#!/dataset/reanalysis-era5-single-levels?tab=form, ECWMF, 2023).

*Author contribution.* HX conceptualized this study. HX, JP and BT carried out the analysis with comments from other co-authors. HX, BT, and JG wrote the paper. JQZ, TC, XG, JZ, and WC provided useful suggestions for the study. JZ contributed to the big data analysis. All authors contributed to the improvement of paper.

*Competing interests.* The contact author has declared that none of the authors has any competing interests.

*Acknowledgments.* This work was jointly supported by the National Natural Science Foundation of China (NSFC) under grants 42205089 and U2142209, the Youth Cross Team Scientific Research Project of the Chinese Academy of Sciences under grant JCTD-2021-10, and the NSFC under grants 41975041 and 41875183.

*Financial support.* This work was jointly supported by the National Natural Science Foundation of China (NSFC) under grants 42205089 and U2142209, the Youth Cross Team Scientific Research Project of the Chinese Academy of Sciences under grant JCTD-2021-10, and the NSFC under grants 41975041 and 41875183.

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

**Tables**

Table 1. The specifications of the Vaisala RS92, Vaisala RS41, and GST1 digital radiosonde.

| Radiosonde characteristics | Vaisala RS92 | Vaisala RS41 | GTS1 digital radiosonde |
|---|---|---|---|
| Manufacturer | Vaisala Oyj, Finland | Vaisala Oyj, Finland | Shanghai Changwang Meteorological Science and Technology Company, China |
| Service period | 2003 to date | 2013 to date | 2002 to date |
| Humidity sensor | Thin-film capacitor, heated twin HUMICAPS | Thin-film capacitor, integrated temperature sensor and heating functionality | Carbon-film hygristor |
| RH range | 0 % to 100 % | 0 % to 100 % | 0 % to 100 % |
| RH uncertainty | 5 % RH | 3 % RH | ~5 % RH |
| Dry bias corrections | Empirical mean bias correction algorithm (Miloshevich et al., 2009); NCAR radiation bias correction algorithm (Wang et al., 2013) | No separate solar radiation correction is needed | Humidity error correction based on fluid dynamic (Mao et al., 2016); PSO-BP neural network correction (Shan et al., 2018) |
| Vertical resolution | 2 s | 2 s | 1 s |
| References | Jauhiainen and Lehmuskero, 2005; Vömel et al., 2007; Miloshevich et al., 2009; Wang et al., 2013 | Jensen et al., 2016; Vaisala, 2017 | Li, 2016; Li et al., 2009; Bian et al., 2011; Chen et al., 2021 |


**Table 2. Summary of altitude-dependent thresholds of RH.**

| Altitude Range | Height-Resolving RH Thresholds | | |
| --- | --- | --- | --- |
| | min-RH | max-RH | inter-RH |
| 0–2 km | 84 % | 94 % | 82 % |
| 2–6 km | 80 % | 92 % | 78 % |
| 6–12 km | 78 % | 88 % | 70 % |
| >12 km | 70 % | 80 % | 70 % |

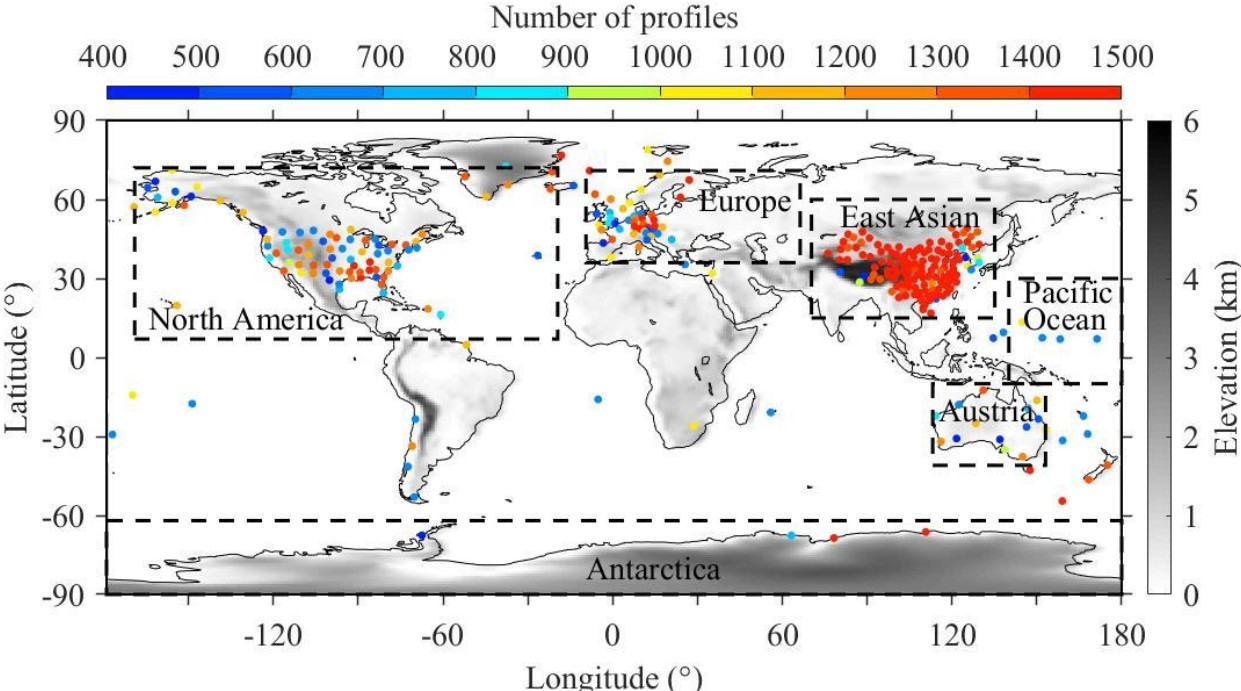

**Figure 1.** Geographic distribution of the number of profiles (colored full circle) for the near-global high-resolution radiosonde observation stations from 2018 to 2019. Also shown is the elevation of each radiosonde site in grey-scale shading. Black rectangles denote observation coverages at six regions of interest (i.e., Europe, North America, East Asia, Austria, Pacific Ocean, and Antarctica). The numbers of radiosonde stations within Europe, North America, East Asia, Austria, Pacific Ocean, and Antarctica are 44, 150, 120, 13, 5, and 4, respectively. The numbers of radiosonde stations in the Northern Hemisphere (NH) and Southern Hemisphere (SH) are 338 and 36, respectively.

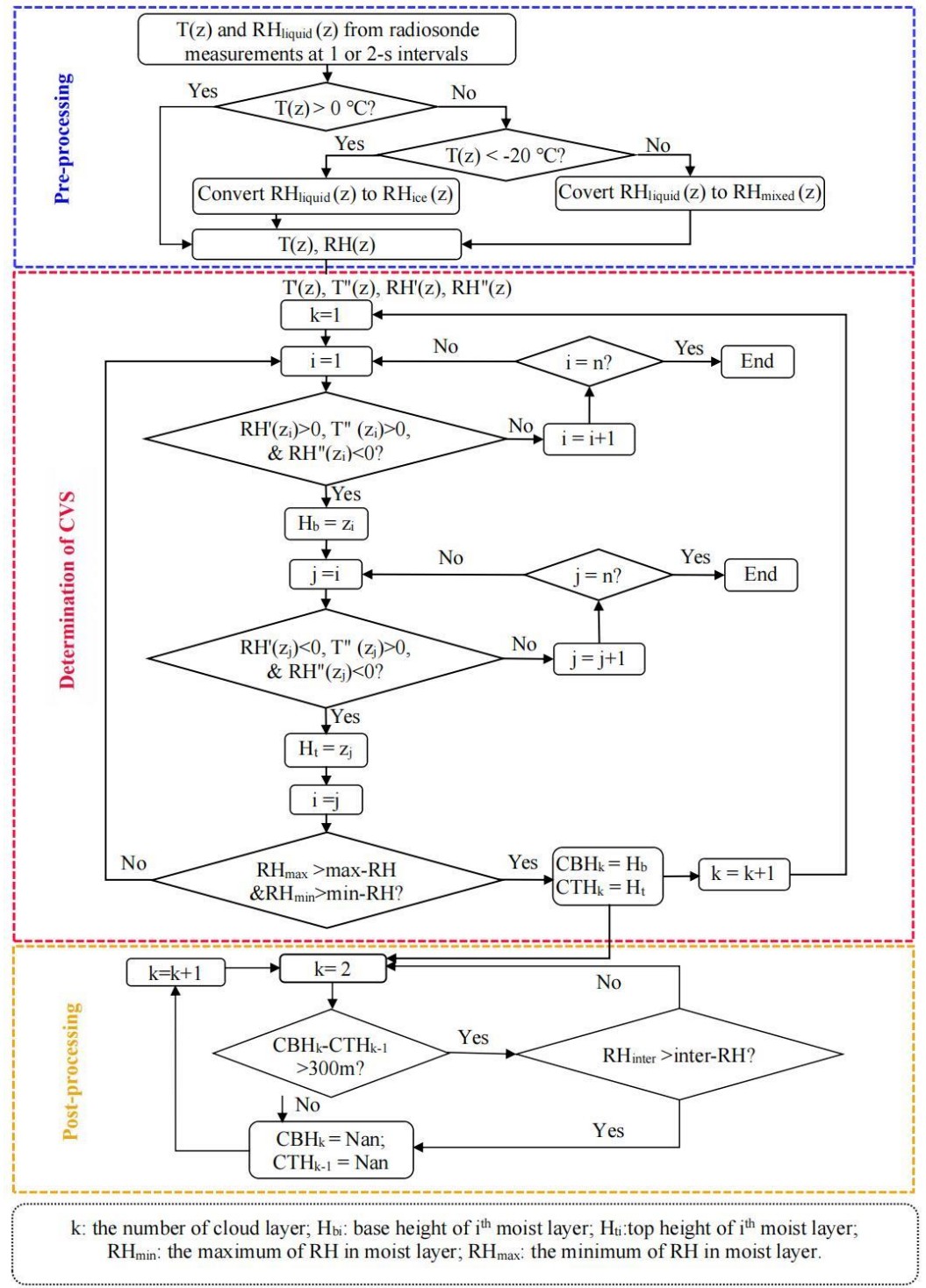

**Figure 2.** Flow chart showing the determination of the cloud vertical structure (CVS) using the vertical profiles of air temperature (T) and relatively humidity (RH) from high-resolution radiosonde data and the altitude-dependent thresholds of RH defined in Table 2.

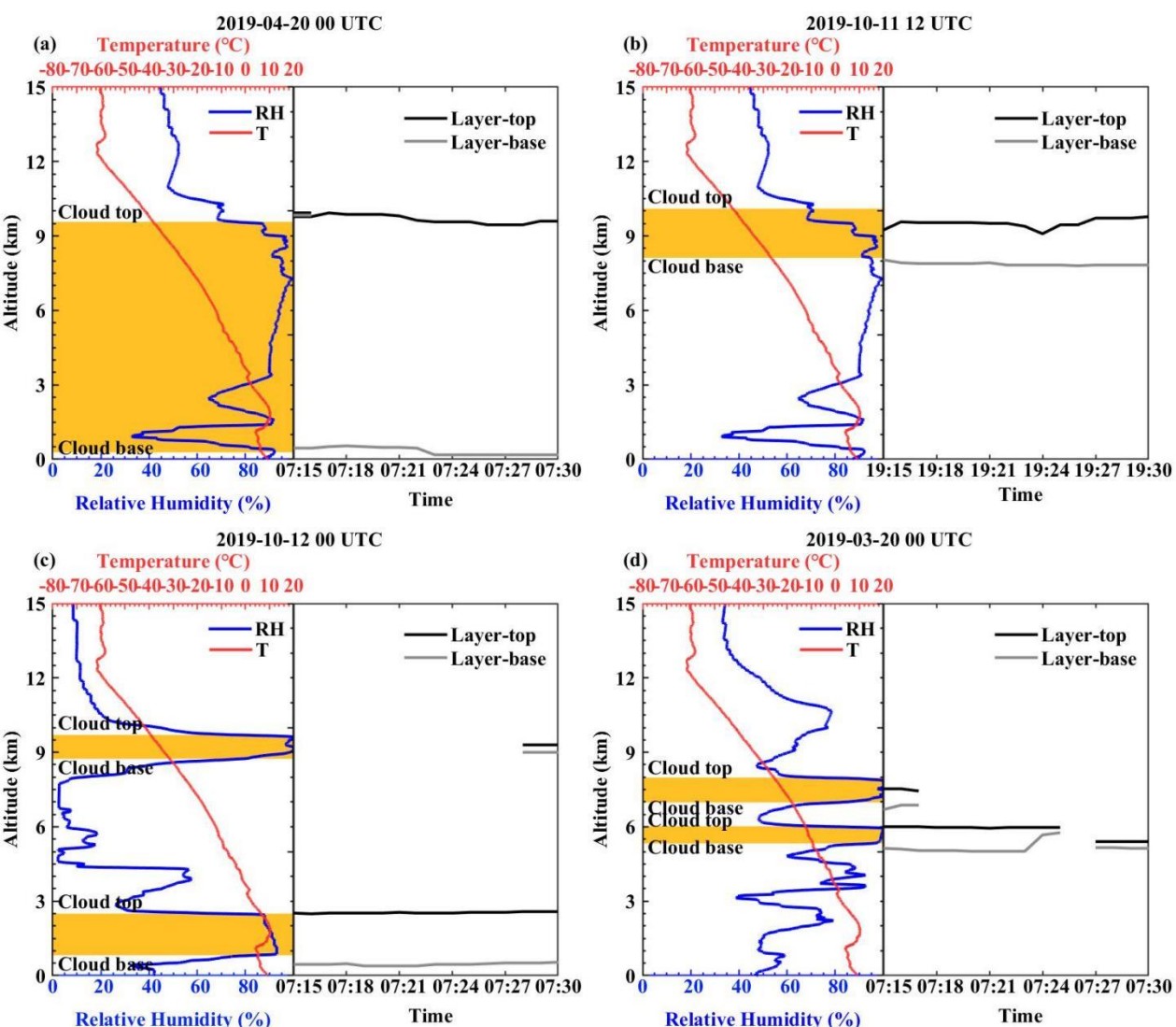

**Figure 3.** Examples of the detection of CVS by (left) high-resolution radiosonde and (right) Ka-band millimeter wavelength cloud radar (MMCR) at Beijing site for the four selected cases, (**a** and **b**) one-layer clouds, and (**c and d**) two-layer clouds. Yellow shading represents the cloud layers retrieved from radiosonde. In each subfigure (left), the blue and red solid line represent the RH and T profile, respectively.

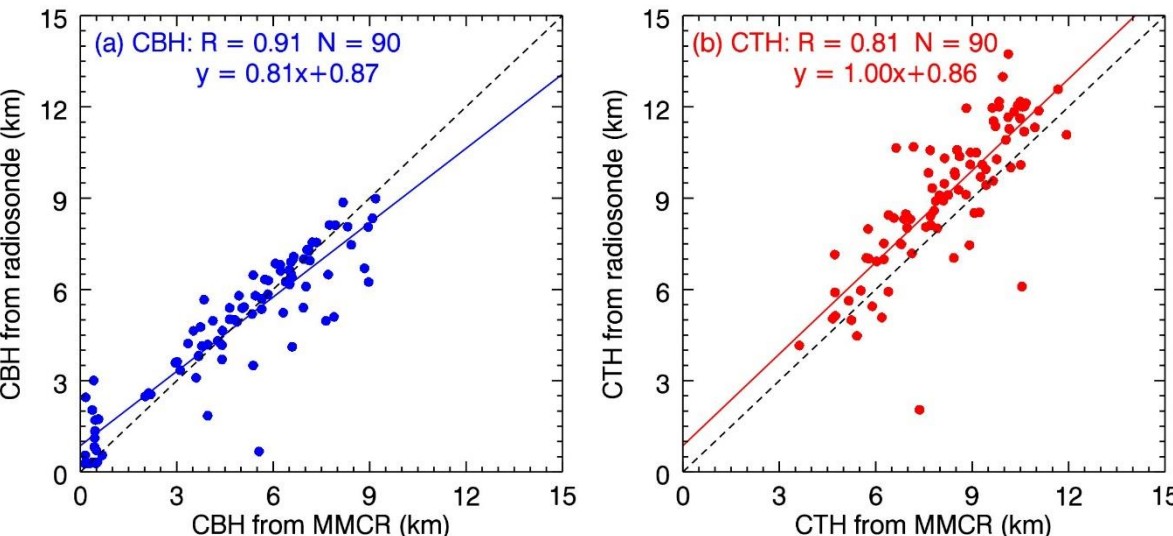

**Figure 4.** Scatter plots of the radiosonde derived (**a**) cloud base heights (CBH) and (**b**) cloud top heights (CTH) versus those from the MMCR at Beijing site during the year of 2019. R represents the correlate coefficient and N represents the sample number. The dashed line and solid line in each panel denote the 1:1 line and the linear regression line, respectively.

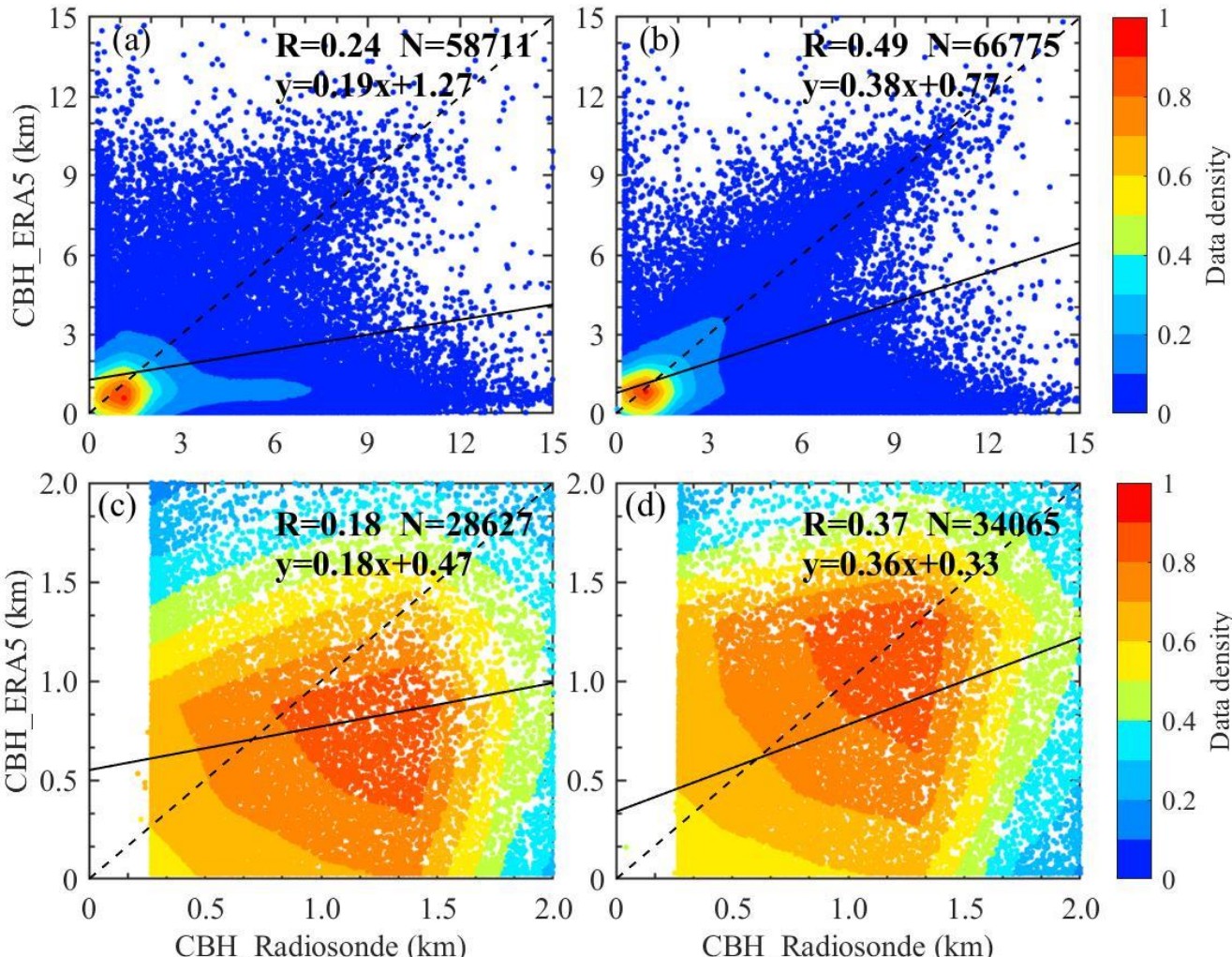

**Figure 5.** Scatter plots of the comparisons of the radiosonde derived CBHs with those from ERA5 for all the data at (**a**) 0000 and (**b**) 1200 UTC, and for the cases that both CBHs are less than 2.0 km at (**c**) 0000 and (**d**) 1200 UTC during the period of 2018–2019. Each point represents one measurement at 0000 or 1200 UTC. R represents the correlate coefficient and N represents the sample number. The linear regression equation is also given, and the regression line is marked in black solid line. The dash black line is 1:1 line.

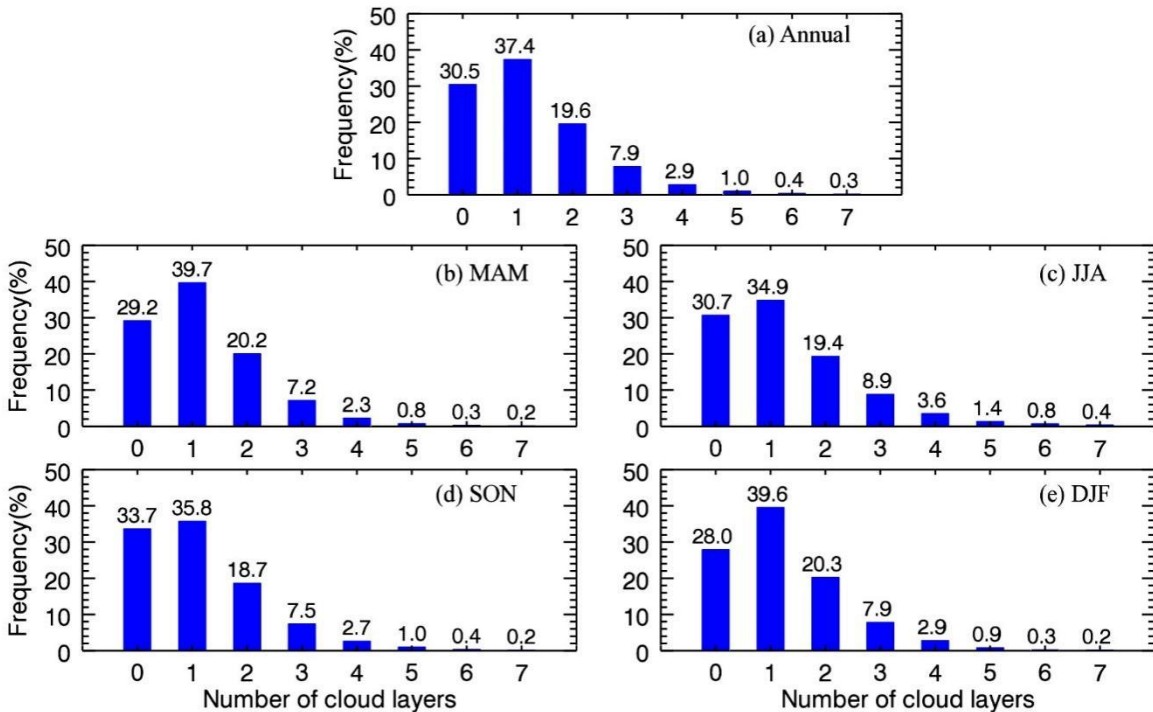

**Figure 6.** Near-global mean occurrence frequencies of clouds with a variety of number of layers ranging from 0 to 7 as detected by high-resolution radiosonde measurements at 1200 UTC during the period of 2018–2019: (**a**) annual, (**b**) March–April–May (MAM), (**c**) June–July–August (JJA), (**d**) September–October–November (SON), and (**e**) December–January–February (DJF). Also marked is the probability for the specified cloud type at the top of each bar.

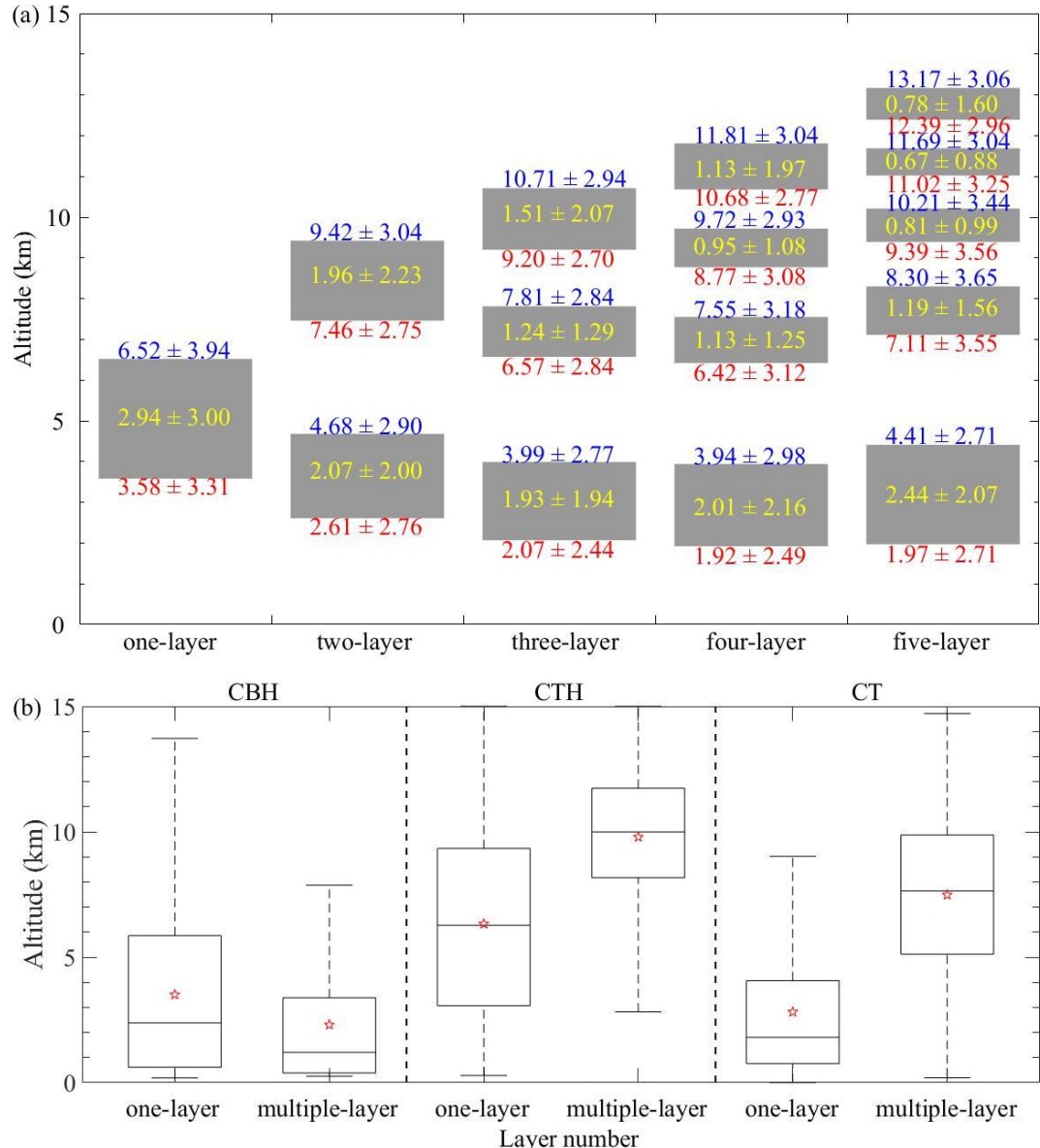

**Figure 7.** Near-global annual mean (**a**) vertical locations of one-, two-, three-, four-, and five-layer clouds and (**b**) boxplot of CVS (CBH, CTH, and CT) for one- and multi-layer clouds at 1200 UTC during the period of 2018–2019. The mean ± one standard deviation values of CBH, CTH, and total CT for each cloud type are also marked in (**a**).

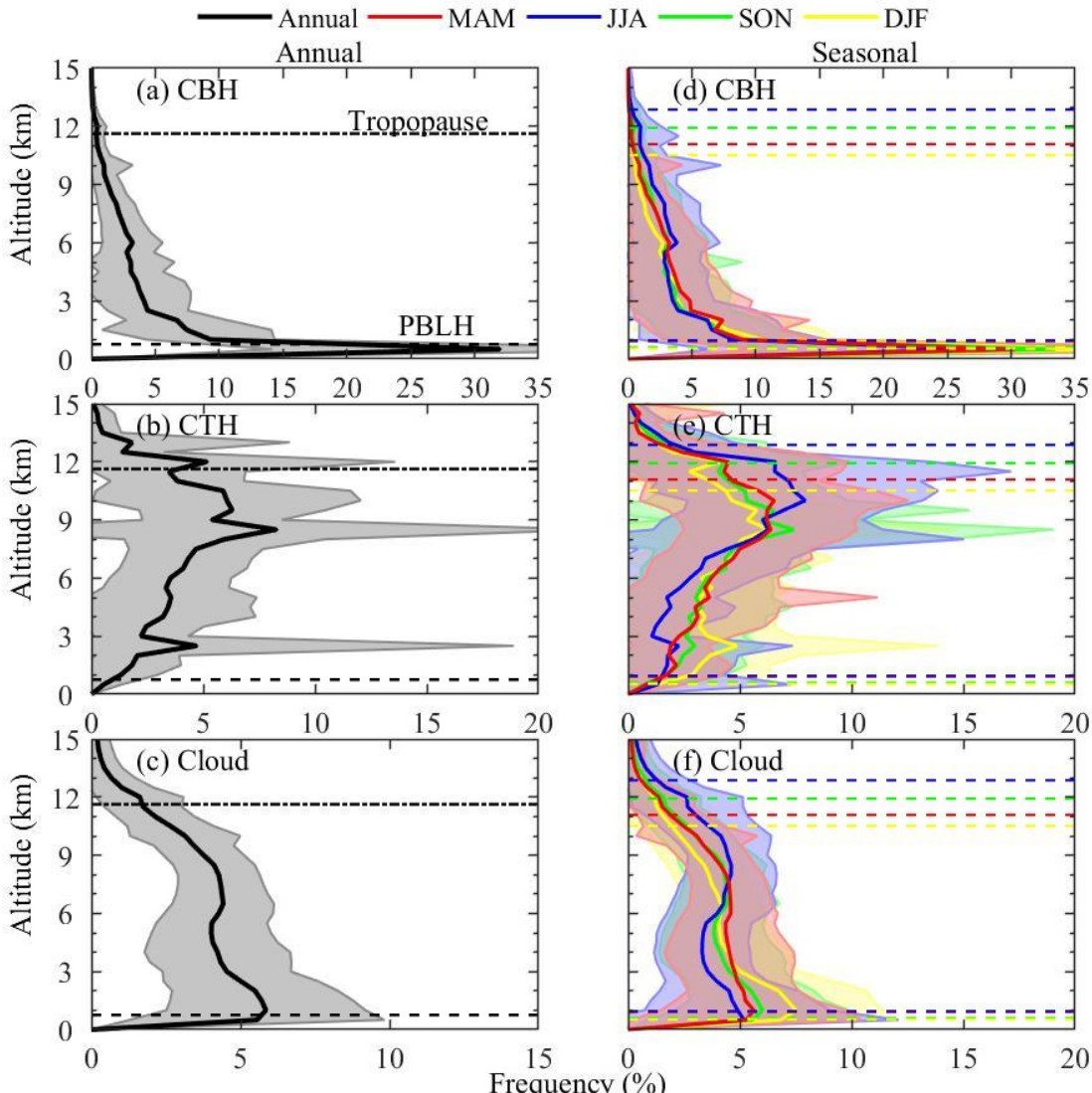

**Figure 8.** Near-global mean vertical distributions of (**a**, **b**, and **c**) annual and (**d**, **e**, and **f**) seasonal occurrence frequencies of CBHs, CTHs, and clouds as detected by radiosonde data at 1200 UTC during the period of 2018–2019, respectively. The annual, MAM, JJA, SON, and DJF are marked in black, red, blue, green, and yellow, respectively. Samples are vertically divided with a resolution of 500 m. The percentage for a given altitude is defined as the ratio of cloudy samples on that altitude to all cloudy samples. The solid lines are the mean values and

shadows are the one standard deviation at annual or a given season. The planetary boundary layer height (PBLH) is determined with the method proposed by Vogelezang and Holtslag (1996), marked in dot-hyphen, and the tropopause is defined with the method from WMO (1957), marked in hyphen. The determination of PBLH and tropopause are detailed in the Supplementary Information.

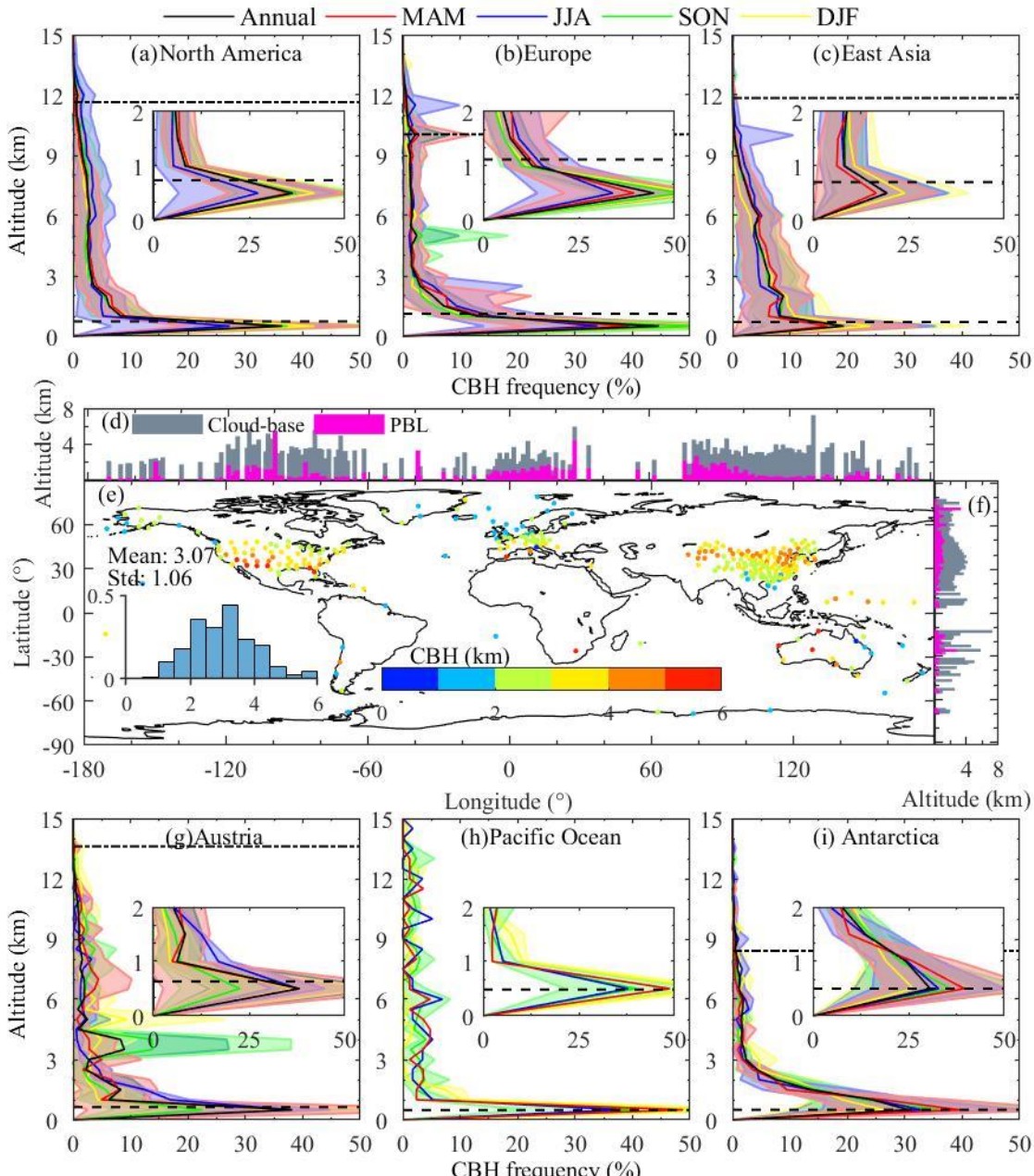

**Figure 9.** Regional mean vertical distributions of the occurrence frequencies of CBHs at 1200 UTC during the period of 2018–2019. The altitude resolved annual and seasonal averaged occurrence frequencies of CBHs are displayed in (**a**, **b**, **c**, **g**, **h**, and **i**) over six regions of interest, including North America, Europe, East Asia, Austria, Pacific Ocean, Antarctica. Also shown are the near-global geographic distribution of the annual mean CBH (**e**), with the histogram of the probability distribution for CBH in the inset and the corresponding meridional (**d**) and zonal (**f**) means overlaid with the mean PBLH.

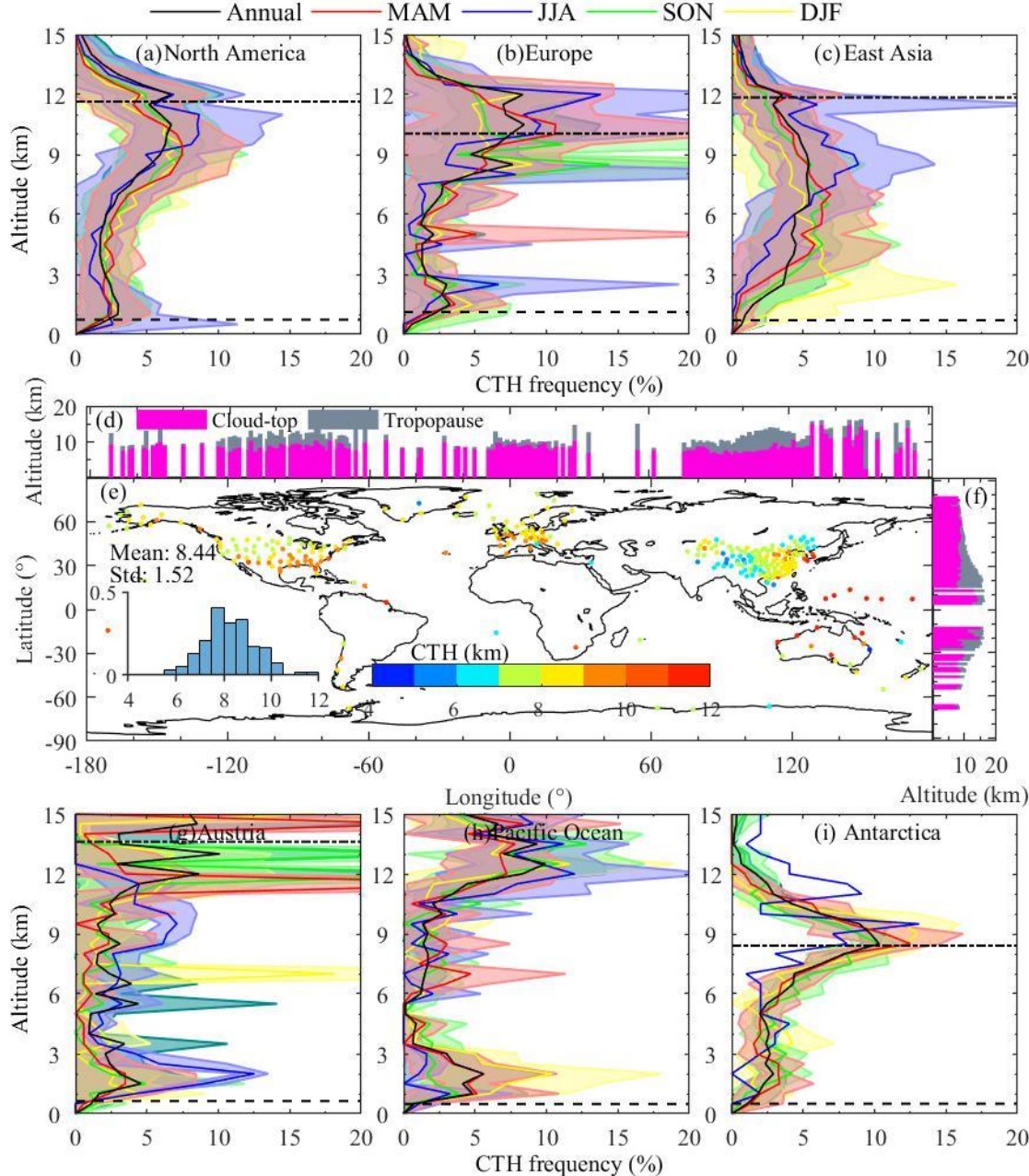

**Figure 10.** Similar as Fig. 9, but for the occurrence frequencies of CTHs at 1200 UTC during the period of 2018–2019.

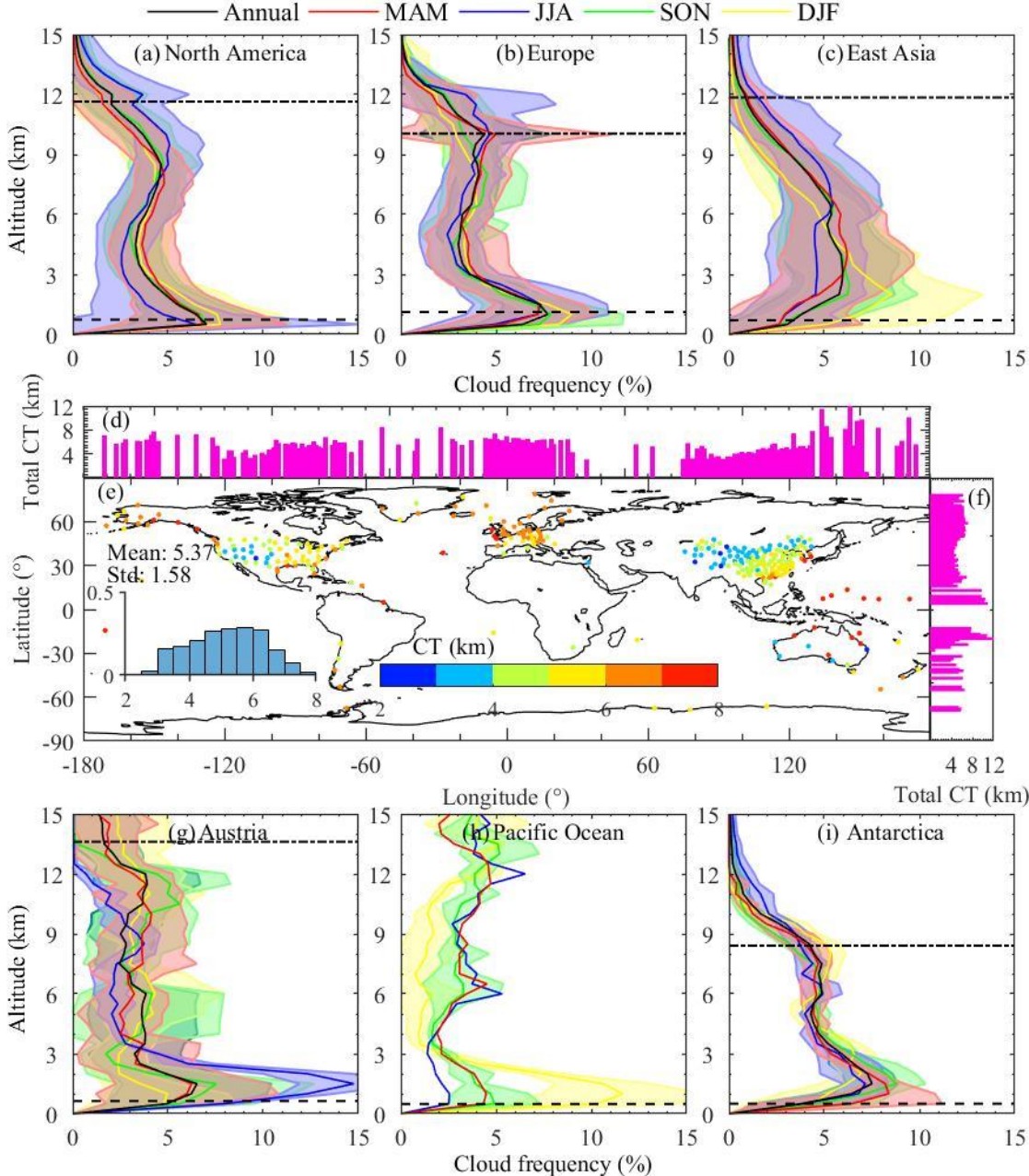

**Figure 11.** Similar as Fig. 9, but for the occurrence frequencies of clouds at 1200 UTC during the period of 2018–2019.

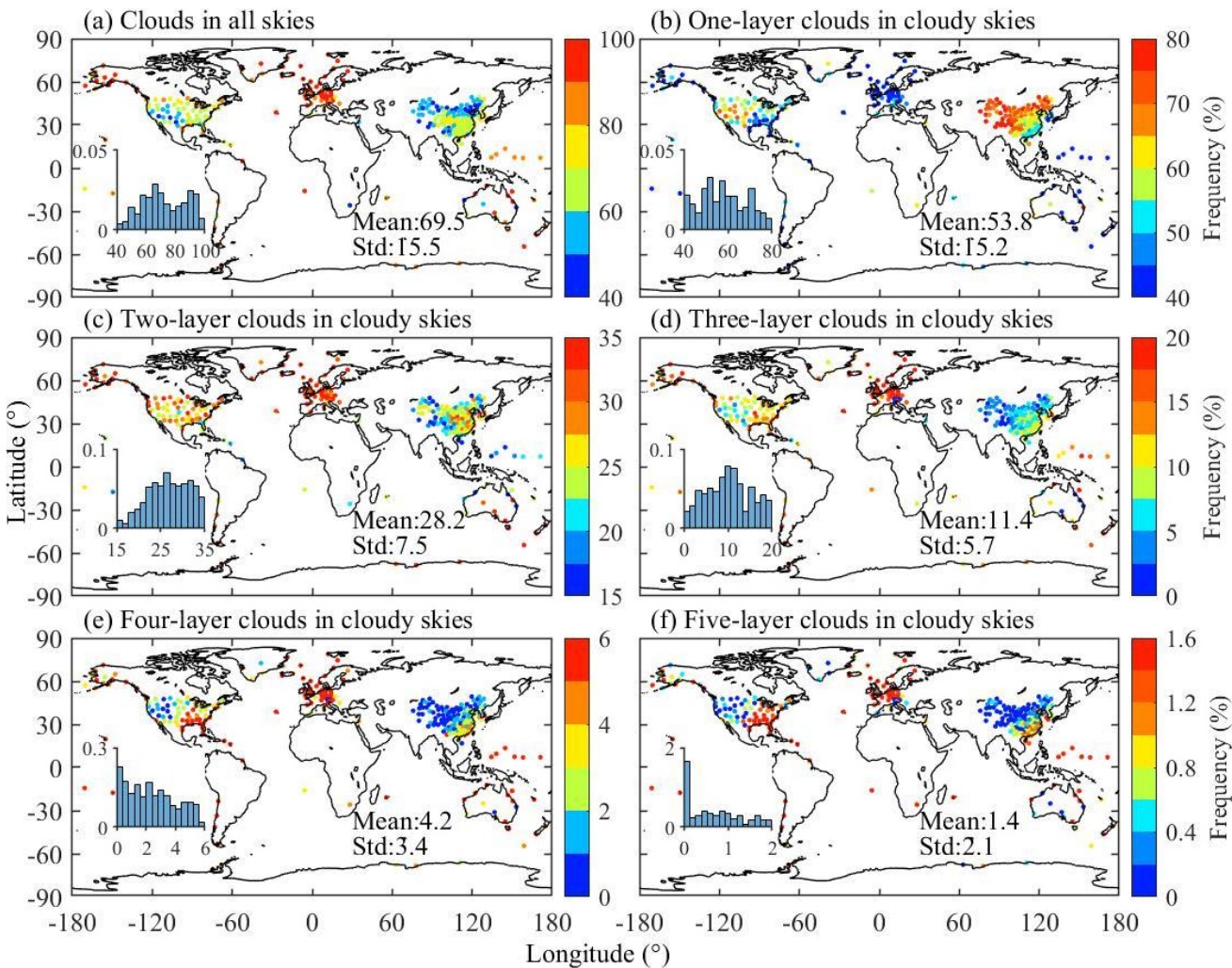


**Figure 12.** The geographic distributions of the occurrence frequencies of (**a**) clouds in all skies and (**b, c, d, e,** and **f**) one-, two-, three, four-, and five-layer clouds in cloudy skies at 1200 UTC during the period of 2018–2019. It should be noted that the range of the color bar differ a lot in order to improve the visual interpretation. Also shown are the histograms of probability distributions for the cloud occurrence

frequencies in each panel.

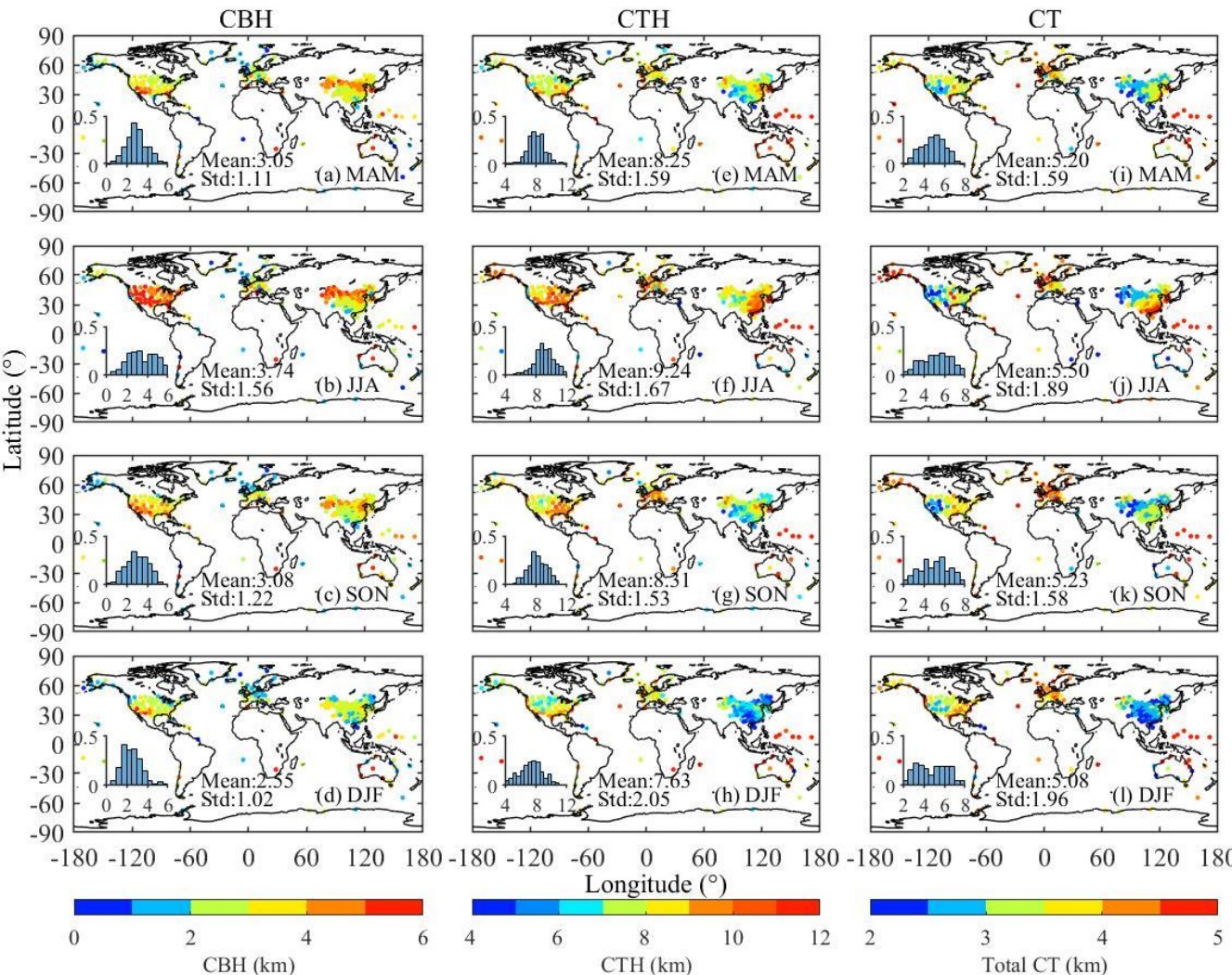

**Figure 13.** The geographic distributions of the seasonal mean CBH (**a**, **b**, **c**, and **d**), CTH (**e**, **f**, **g**, and **h**), and CT (**i**, **j**, **k**, and **l**) at 1200 UTC during the period of 2018–2019. Also shown are the histograms of probability distributions for the CVS in each panel.


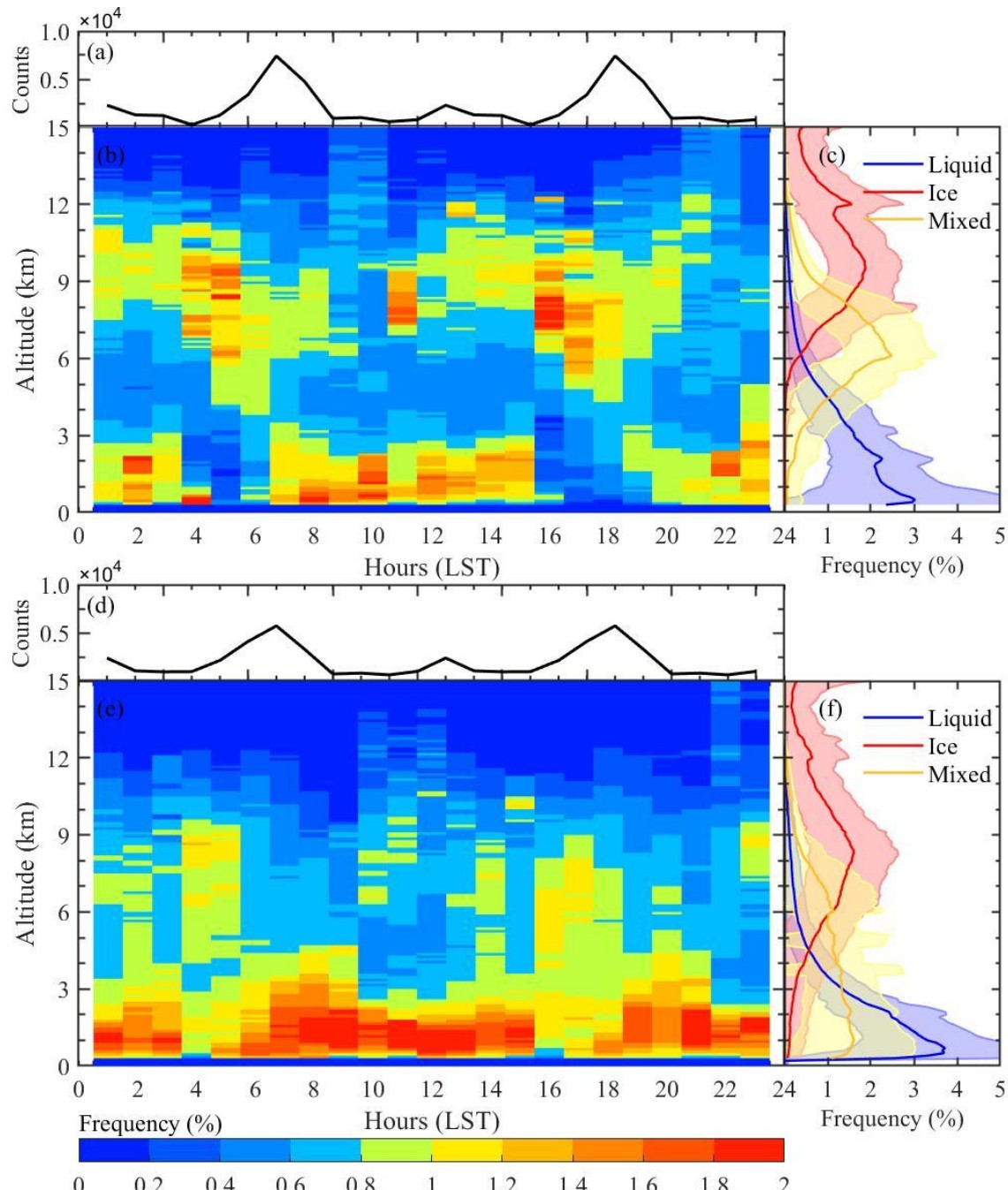

**Figure 14.** Diurnal variations of all clouds and altitude-resolved clouds as detected by all radiosondes at 0000 and 1200 UTC in boreal summer (JJA; **a** and **b**) and boreal winter (DJF; **d** and **e**) during the period of 2018–2019. Also shown are the vertical probability distributions of liquid (in blue), ice (in red), and liquid-ice mixed (in yellow) clouds in boreal summer and boreal winter (**c and f**).

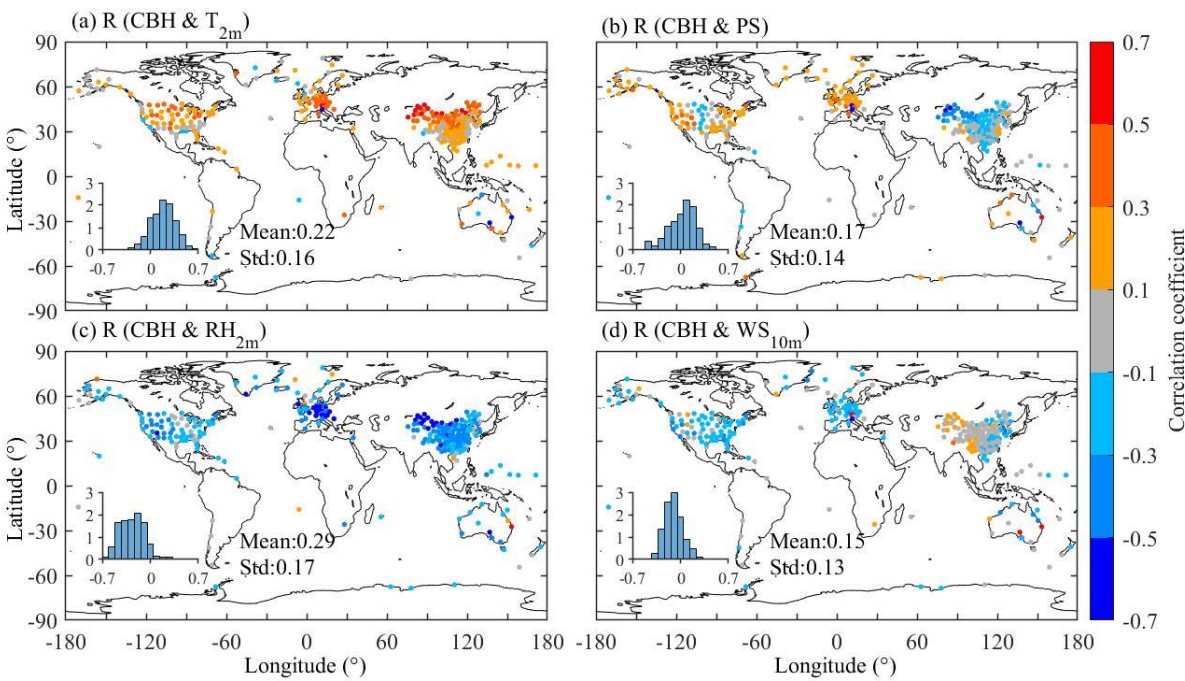

**Figure 15.** Geographic distributions of the correlation coefficients (Rs) between radiosonde derived CBH and surface meteorological
variables: (**a**) 2m air temperature ($T_{2m}$), (**b**) surface pressure (PS), (**c**) 2m relatively humidity ($RH_{2m}$), and (**d**) 10m wind speed ($WS_{10m}$) at
1200 UTC during the period of 2018–2019. Also shown are the histograms of probability distributions for their corresponding R values in
each panel.

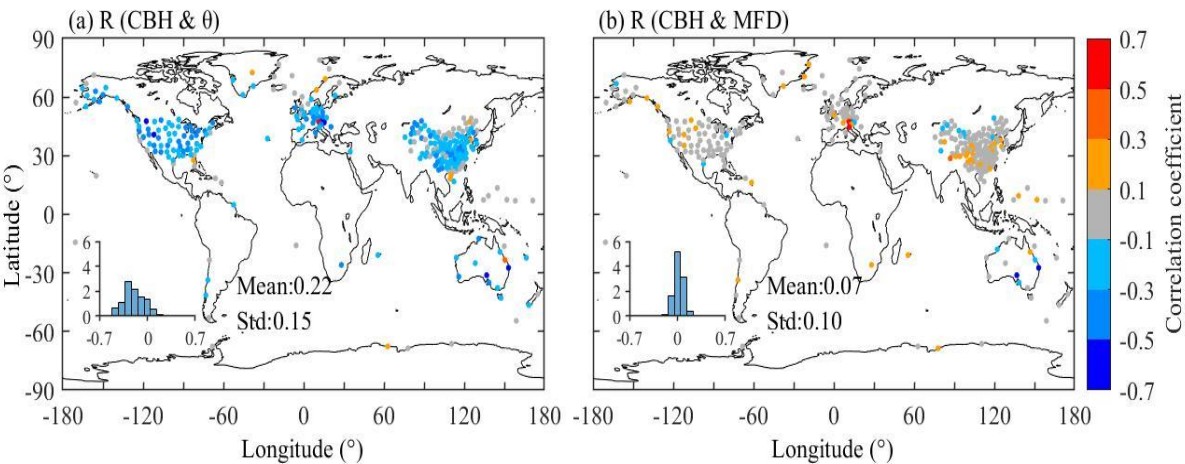

**Figure 16.** The same as Fig. 15, but for the correlations between CBH and (**a**) soil water content ($\theta$) and (**b**) moist flux divergence (MFD) at 1200 UTC during the period of 2018–2019.

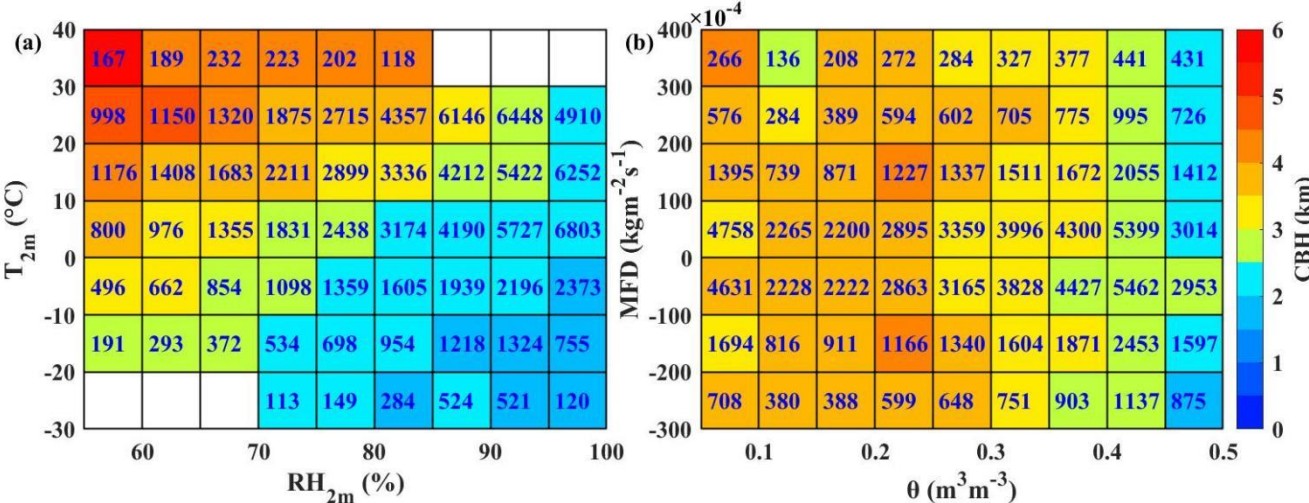

**Figure 17.** Joint dependences of CBH on (**a**) $T_{2m}$ and $RH_{2m}$, (**b**) $\theta$ and MFD at 1200 UTC during the period of 2018–2019. The number labelled in each cell represents its corresponding sample size.