# Peer review of "Characterizing the near-global cloud vertical structures over land using high-resolution radiosonde measurements"

_EGUsphere, 2023_

## Author Comment (AC1)

**Response to Reviewer 1' Comments**

Comments on "Characterizing the near-global cloud vertical structures over land using high-resolution radiosonde measurements"

**General Comments**

This paper examines the near-global cloud vertical structures using two years of radiosonde data. I do not find any major flaws with their methodology and conclusions, and the statistical results could be a nice contribution to modeling global cloud radiative effects. However, clarifications are needed to make this paper a compelling story. I suggest returning to the authors for minor revision.

Response: We thank the reviewer for his/her comprehensive evaluation and thoughtful comments, which help tremendously to improve the quality of our work. We have tried our best to address the reviewer' concerns one by one. For clarity purpose, here we have listed the reviewer' comments in black, followed by our responses in blue, and the modifications to the manuscript are in italics. We sincerely hope that the reply and the revisions can satisfy the editor and referee' expectations.

**Major Comments**

The Introduction section listed several previous works using lowering resolution radiosonde data to retrieval cloud boundaries but did not include a summary of what were found from those works, what are the main statistical and conclusions from those works. Most importantly, the authors should articular what are novel in the current study, in addition to higher resolution data.

Response: Per your kind suggestions, we added the main conclusions of previous studies in the revised version (Manuscript_tracked.docx) as follows:

"Poore et al. (1995) proposed a T-dependent dewpoint depression threshold for cloud detection, *and they found that only high clouds exhibited strong latitudinal and seasonal variation in the thickness of cloud layer*. Wang and Rossow (1995) detected cloud layers using single RH threshold, with the maximum and minimum RH thresholds of 87 % and 84 %, respectively. *They demonstrated that the occurrence frequency of multi-layer clouds varied geographically and multi-layer clouds occurred most frequently in the tropics.* Zhang et al. (2010) improved the single threshold method by using an altitude-dependent RH thresholds to characterize the base and top of cloud layers, *and they demonstrated that multilayer clouds occurred more frequently in the summer*. Another method is the gradient method, in which cloud layers are obtained by examining the variations of RH and T profiles. Chernykh and Eskridge (1996) used a second derivative of the vertical profiles of RH and T to determine cloud boundaries, *and they indicated that the accuracy of the prediction of cloud level was independent of the level type and location*."

In addition to the higher vertical resolution radiosonde data, a novel CVS detection method is developed in this study. We revised the objective of our study as follows:

"The main objective of present study is to provide the first attempt to retrieve near-global vertical structures of clouds, including the number of cloud layers, cloud base height (CBH), cloud top height (CTH), and cloud thickness (CT) of each layer, using two years' worth (2018–2019) of high-vertical-resolution (5–10 m) radiosonde observations from 374 radiosonde stations across the world. *In order to obtain better CVS results, we first develop a novel CVS detection method that integrates the two main methods mentioned above by considering both the vertical gradients of RH and T, as well as the altitude-dependent thresholds of RH*".

**Minor Comments**

1.Line 17-19: cloud base height correlate with millimeter wavelength radar?
Response: To clarify the sentence, we changed the sentence "It is found that the cloud base heights (CBHs) from the radiosondes have a higher correlation coefficient (R = 0.91) with the millimeter wavelength cloud radar than that with the ERA5 reanalysis (R = 0.49)" to "It is found that the cloud base heights (CBHs) from the radiosondes have a higher correlation coefficient (R = 0.91) with the *CBHs from* millimeter wavelength cloud radar than *those from* the ERA5 reanalysis (R = 0.49)".

2.Line 52: do you mean the Chang and Li retrievals have large discrepancies? Discrepancies relative to what?
Response: Thanks for pointing out this mistake. Chang and Li (2005) have obtained reliable near-global CVS for one-layer and overlapped clouds by applying a new method to MODIS data. Their retrievals can differentiate cirrus overlapping lower water clouds from single-layer clouds, but cannot provide the vertical structures of three- or more-layer clouds. We changed the sentence "However, these retrievals existed large discrepancies, especially for high cirrus overlapping lower water clouds" to "However, these retrievals *lack the vertical structures of three- or more-layer clouds*" in the revised version (Manuscript_tracked.docx).

3.Line 55-56: the last sentence needs to be revised. Polar orbiting satellites can have short revisit periods such as AQUA/TERRA. Do you mean 'narrower nadir views' ?
Response: Thanks for your reminder. The sentence was revised as "*Active sensors have relatively long revisit periods (e.g., 16-day) and narrow nadir views (e.g., Winker et al., 2007; Kim et al., 2011; Guo et al., 2016)*" in the revised version (Manuscript_tracked.docx).

4.Line 58: cloud radars
Response: As suggested, we changed "cloud radar" in Line 58 to "*cloud radars*".

5.Line 75: do you mean the vertical resolution, horizontal resolution, or temporal resolution?

Response: Here, the resolution refers to the vertical resolution. We changed the "resolution" as "*vertical* resolution" in the revised version (Manuscript_tracked.docx).

6.Line 75-79: it will be more intuitive to understand the difference of 'resolution' (whatever it refers to) from previous and current radiosondes if you can provide several numbers here.

Response: Thanks for your great suggestions. We provided the specific value of the resolution for radiosonde used in previous and current studies in the revised version (Manuscript_tracked.docx), as follows:

"The possible reasons can be concluded as (1) the vertical resolution of atmospheric profiles provided by radiosonde is low *(e.g., 76 meters (m); Poore et al., 1995)*, and (2) refined RH thresholds remain lacking for cloud detection".

"With the emergence of growing number of high-vertical-resolution *(5–10 m)* radiosonde measurements worldwide, improved retrievals of CVS on large scale are now plausible".

7.Line 107: change 'considered' to 'included'

Response: Corrected as suggested.

8.Line 115: an accuracy of

Response: Corrected as suggested.

9.Line 124-125: references for the ERA5 reanalysis are needed here

Response: Done. The reference "*(Bell et al., 2021)*" for the ERA5 reanalysis was listed in the References Section as follows:

*Bell, B., Hersbach, H., Simmons, A., Berrisford, P., Dahlgren, P., Horanyi, A., Munoz-Sabater, J., Nicolas, J., Radu, R., Schepers, D., Soci, C., Villaume, S., Bidlot, J. R., Haimberger, L., Woollen, J., Buontempo, C., and Thepaut, J. N.: The ERA5 global reanalysis: Preliminary extension to 1950, Q. J. Roy. Meteor. Soc., 147, 4186–4227, https://doi.org/10.1002/qj.4174, 2021.*

10.Line 168: enters a moist layer

Response: Corrected as suggested.

11.Line 190: can you explain why a max-RH is needed to detect a cloud layer? What is inter-RH in Table 1 and Figure 2? Is it the RH between consecutive cloud layers?

Response: In the detection of cloud layer, some thin moist layer could be recognized to be cloud layer. Therefore, as previous studies (e.g., Wang and Rossow, 1995; Zhang et al., 2010), we used a max-RH to minimize this issue. To clarify the reason for using max-RH to detect a cloud layer, we added the sentence "*By using max-RH, it is possible to avoid misidentifying some thin moist layer as cloud layer*" in line 207 of the revised version (Manuscript_tracked.docx).

In Table 1 and Figure 2, the inter-RH is the minimum RH thresholds between the consecutive cloud layers. We changed the description "within this distance" to

"*between the consecutive cloud layers*" in line 216 of the revised version (Manuscript_tracked.docx).

12.Line 184-191: do you do any averaging or smoothing for the RH and T profiles, considering they are in high vertical resolution?
Response: Yes, we did average for the RH and T profiles before determining the CVS. Additional text was added at the end of Section 2.2.1 of the revised version (Manuscript_tracked.docx), as follows:

"*Before determination of CVS, the profiles of $RH(z)$ and $T(z)$ after the above pre-processing are smoothed by the arithmetical averages of $RH(z)$ and $T(z)$ at the altitudes of $z_{i-1}$, $z$, and $z_{i+1}$ ($i \geq 2$), respectively.*"

13.Figure 3: I suggest change sounding times to 00UTC and 12UTC to be consistent with your intro text
Response: Thanks for your reminder. Here, we revised the 2300 UTC and 1100 UTC to 0000 UTC and 1200 UTC, respectively, in Figure 3, as follows:

[Figure]

**Figure 3**. Examples of the detection of CVS by (left) high-resolution radiosonde and (right) Ka-band millimeter wavelength cloud radar (MMCR) at Beijing site for the four selected cases, (a, b) one-layer clouds, and (c, d) two-layer clouds. Green shading represents the cloud layers retrieved from radiosonde. In each subfigure (left), the blue and red solid line represent the RH and T profile, respectively.

14.line 223: maybe change the word 'correctly' to 'reasonably'
Response: Thanks. We changed "correctly" to "*accurately and reasonably*".

15.line 313-314: these result in the occurrence
Response: Done.

16.line 368: oceanic climate
Response: Done.

---

## Author Comment (AC2)

**Response to Reviewer 2' Comments**

Review of "Characterizing the near-global cloud vertical structures over land using high-resolution radiosonde measurements" by Xu et al., for publication in EGUsphere

**General Comments**

The main point of this manuscript examines cloud vertical structure using radiosonde data from 374 land stations. Millimeter wavelength radar estimated cloud boundaries have a high correlation to radiosondes relative to ERA-5 derived cloud vertical structure, which is unsurprising. This study analyzes multi-layer clouds, with their analysis noting several instances of 3 or more cloud layers measured by a single radiosonde. This study is packed with interesting information about global cloud statistics, particularly how they vary in different regions of the world and for liquid, mixed and ice phase clouds. Their results discussing seasonal cloud boundaries are in very good agreement with several previous studies also using radiosondes for cloud property measurements. The figures are very high quality and complement the text very well.

There are few areas where this manuscript needs improvement. First, there is very little discussion about the radiosonde types or any discussion of measurement calibration/uncertainty. This is extremely important given the volume of radiosondes and noting that different versions (e.g., the Vaisala RS41 and RS92) were developed differently. The Vaisala RS92 in particular is prone to an RH dry bias, and there is no mention if those sondes (if they were used at all) employed any sort of correction or homogenization to the global database (aside from what we know about the GRUAN database). The authors need to make these points much more clear and do a better job of convincing the reader that the measurements are indeed homogenized. I think this can be accomplished in 1-2 additional paragraphs in the methods section, along with a table highlighting manufacturer/temperature/humidity (etc.) uncertainty and accuracy, along with documented studies noting any biases. Second, I think the authors missed a fantastic opportunity to explore their results in the context of relative humidity with respect to ice or RH(ice). RH(ice) is key for ice cloud formation, and though there are many studies that caution against the use of radiosonde relative humidity especially at high altitudes, the statistics of RH(ice) would be interesting to present nonetheless as it would give clear indication which climates around the world are most conducive to ice supersaturation. If the authors choose to add this to the paper, they will need to also ensure the uncertainty is well documented. In addition, there are several technical, grammatical and spelling errors in this manuscript that – while not significant in volume – was distracting and made the paper hard to read at times. I encourage the readers to carefully check their work for these errors.

Overall, this paper is a very extensive analysis of global cloud coverage that fits well within the scope of EGUsphere, and should be considered for publication after addressing several comments below.

Response: We thank the reviewer for his/her comprehensive evaluation and thoughtful comments, which help tremendously to improve the quality of our work. We have tried our best to address the reviewer' concerns one by one. For clarity purpose, here we have listed the reviewer' comments in black, followed by our responses in blue, and the modifications to the manuscript are in italics. We sincerely hope that the reply and the revisions can satisfy the editor and referee' expectations.

**Specific Comments**

1. L17: It would be good to elaborate a bit here in the abstract where these 374 land stations are partitioned.

Response: Thanks to your good suggestion, we changed the sentence " In this research, near-global CVS is characterized by high-vertical-resolution twice daily radiosonde observations from 374 stations over land." to "In this study, near-global CVS is characterized by high-vertical-resolution twice daily radiosonde observations from 374 stations over land, *which distributed in Europe, North America, East Asia, Austria, Pacific Ocean, and Antarctica*." in the revised version (Manuscript_tracked.docx).

2. L37-48: This is a solid introductory motivation.

Response: Thank you very much for your recognition.

3. L57: This is a bit awkwardly written. Perhaps consider moving the Hahn et al. (2001) reference to the end of the sentence.

Response: As suggested, the sentence was revised as "Ground-based instruments, such as lidars (Gouveia et al., 2017), ceilometers (Costa-Surós et al., 2013), and cloud radars (Mace et al., 1998), have proven to be effective in providing CVS with continuous temporal coverage and relatively high accuracy (*Hahn et al., 2001*; Zhou et al., 2020)." in the revised version (Manuscript_tracked.docx).

4. L57-61: I would be careful making the assertion that coverage of these ground-based radars/lidars/ceilometers are limited to "a few locations". You should expand this paragraph by at least 2-3 sentences and highlight where these locations are, and demonstrate to the reader that these measurements are indeed few. Otherwise, it undermines (in my opinion) a big part of the motivation of this research. The Atmospheric Radiation Measurement (ARM) program has many of these sites listed and available, and are definitely more than a few.

North Slope Alaska:

[revised manuscript text omitted]

5. L72-76: You should review the "cirrus cloud detection algorithm" subsection in Dzambo and Turner (2016) as their method provided a viable radiosonde/ground-based radar/lidar collocation algorithm. Their method was by no means perfect, but their method established both spatial and temporal restrictions to ensure a radiosonde was indeed launched into a cloud.

Dzambo, A. M., and Turner, D. D. (2016), Characterizing relative humidity with respect to ice in midlatitude cirrus clouds as a function of atmospheric state, J. Geophys. Res. Atmos., 121, 12,253– 12,269, doi:10.1002/2015JD024643.

Also consider the role of their "lag time" correction, which is also in this section.

Response: As suggested, we added the sentence "*Dzambo and Turner (2016) identified cirrus based on a cirrus cloud detection algorithm by using radiosonde and cloud radar data and found that RH with respect to ice within cirrus clouds varied seasonally, with maximum in winter and minimum in summer. To ensure the radiosonde measurements were collocated with the appropriate MMCR measurements, they established temporal ("lag time") and spatial restrictions.*"

References:

*Dzambo, A. M., and Turner, D. D.: Characterizing relative humidity with respect to ice in midlatitude cirrus clouds as a function of atmospheric state, J. Geophys. Res.-Atmos., 121, 12253–12269, https://doi.org/10.1002/2015JD024643, 2016.*

6. L82: Where in the world are these 374 land stations? A few examples would be good to note here for the reader.

Response: As suggested, we added "*(e.g., Europe, North America, East Asia, Austria, Pacific Ocean, and Antarctica)*" to the sentence "The main objective of present study is to provide the first attempt to retrieve near-global vertical structures of clouds, including the number of cloud layers, cloud base height (CBH), cloud top height (CTH), and cloud thickness (CT) of each layer, using two years' worth (2018–2019) of high-vertical-resolution (5–10 m) radiosonde observations from 374 radiosonde stations across the world *(e.g., Europe, North America, East Asia, Austria, Pacific Ocean, and Antarctica)*."

7. L90: technical correction: "... we also investigate the relationship between CBH, surface meteorology, and moisture."
Response: Corrected as suggested.

8. Section beginning at L95: There is a very important piece of information missing from this section... the manufacturing information of all radiosondes used in your database. Were these radiosondes Vaisala RS-92? Vaisala RS-41? Because your results are very sensitive to the relative humidity measurements of the radiosonde, it is also necessary to know what humidity sensors are on each radiosonde, and by extension, it is further necessary to know and understand the relative humidity uncertainty with each. There are numerous studies discussing the topic about relative humidity corrections in radiosondes. Vaisala RS-92 radiosondes have a very well documented dry bias in their measurements, and to the extent of my knowledge, only the GRUAN database of radiosondes have their humidity products homogenized between different versions. I strongly recommend updating this section of the paper with at least a paragraph discussing the humidity measurements, as well as adding a table of the different sensors from each manufacturer, perhaps something like: manufacturer, years used, reference for sensor, instrument uncertainty, and (if applicable) known biases and corrections such as those for the RS-92.
Wang, J., L. Zhang, A. Dai, F. Immler, M. Sommer, and H. Vömel, 2013: Radiation Dry Bias Correction of Vaisala RS92 Humidity Data and Its Impacts on Historical Radiosonde Data. J. Atmos. Oceanic Technol., 30, 197–214, https://doi.org/10.1175/JTECH-D-12-00113.1.
Miloshevich, L. M., Vömel, H., Whiteman, D. N., and Leblanc, T. (2009), Accuracy assessment and correction of Vaisala RS92 radiosonde water vapor measurements, J. Geophys. Res., 114, D11305, doi:10.1029/2008JD011565.
Vömel, H., and Coauthors, 2007: Radiation Dry Bias of the Vaisala RS92 Humidity Sensor. J. Atmos. Oceanic Technol., 24, 953–963, https://doi.org/10.1175/JTECH2019.1.
Dzambo, A. M., Turner, D. D., and Mlawer, E. J.: Evaluation of two Vaisala RS92 radiosonde solar radiative dry bias correction algorithms, Atmos. Meas. Tech., 9, 1613–1626, https://doi.org/10.5194/amt-9-1613-2016, 2016.
Jensen, M. P., Holdridge, D. J., Survo, P., Lehtinen, R., Baxter, S., Toto, T., and Johnson, K. L.: Comparison of Vaisala radiosondes RS41 and RS92 at the ARM

Southern Great Plains site, Atmos. Meas. Tech., 9, 3115–3129, https://doi.org/10.5194/amt-9-3115-2016, 2016.

de Boer, G., Calmer, R., Jozef, G. et al. Observing the Central Arctic Atmosphere and Surface with University of Colorado uncrewed aircraft systems. Sci Data 9, 439 (2022). https://doi.org/10.1038/s41597-022-01526-9 (see methods section of this paper)

These are papers that should provide good context for RS41 and RS92 humidity measurements. As for the other radiosondes that may have been used in your study, please search for and add documentation similar to what these studies have in addressing humidity measurements.

Response: As suggested, we added the additional text as follows:

[revised manuscript text omitted]

9. L125: The comment here about the ERA-Interim is unnecessary.

Response: We deleted "Compared with former ERA-Interim" and revised the sentence as follows:

*"The temporal and spatial resolutions of ERA5 can reach up to 1 hour (h) and 0.25° × 0.25°, respectively (Hersbach et al., 2020)."*

10. L130: The inclusion of soil moisture content as part of your analysis is interesting, but can you provide context (perhaps a reference or two) showing how this ERA-5 variable was used in previous studies (especially for surface latent fluxes, clouds, or something similar).

Response: As suggested, we added these sentences "*As a key variable that links land surface to cloud formation, soil moisture from the ERA5 reanalysis has been widely used in the analysis of land-atmosphere coupling (Sun et al., 2020). By using the ERA5 reanalysis in East Asia, Wei et al. (2021) explored the relationships between soil moisture, land surface sensible and latent heat fluxes, and CBH, and found the negative correlations between soil moisture and CBH.*"

Response: Thanks. We added the result related to subvisible cirrus in the conclusions, as follows:
"*The mean CTHs are highest at tropic western Pacific, where subvisible cirrus mostly occurs.*"

---

## Author Response (AR2)

**Response to Reviewers' Comments**

First of all, we thank the reviewers for their comprehensive evaluations and thoughtful comments, which help tremendously to improve the quality of our work. We have tried our best to address the reviewers' concerns one by one. For clarity purpose, here we have listed the reviewers' comments in black, followed by our responses in blue, and the modifications to the manuscript are in italics. We sincerely hope that the reply and the revisions can meet the expectations of editor and reviewers.

**Reviewer #1:**

Comments on "Characterizing the near-global cloud vertical structures over land using high-resolution radiosonde measurements" by Xu et al.

**General Comments**

My comments and questions have been adequately addressed in the revision and I found the manuscript in good shape. I recommend this manuscript to be accepted for publication in its current form.

Response: We deeply appreciate the reviewer's approval of our last revised manuscript. Thanks again for the thoughtful comments, which help tremendously to improve the quality of our manuscript.

**Reviewer #2:**

Comments on "Characterizing the near-global cloud vertical structures over land using high-resolution radiosonde measurements" by Xu et al.

**General Comments**

The current version of the manuscript, pending a couple of extremely minor and quick technical/grammar corrections, is ready for full publication in EGUsphere.

Response: We deeply appreciate the reviewer's approval of our last revised manuscript. According to EGUsphere's requirements, we have tried our best to carefully review the technical/grammatical corrections of the manuscript in the revised version (Manuscript_tracked.docx).

**Reviewer #3:**

Comments on "Characterizing the near-global cloud vertical structures over land using high-resolution radiosonde measurements" by Xu et al.

**General Comments**

This study explores the cloud vertical structure using twice daily radiosonde observations. The main focused cloud properties are on the number of cloud layers, cloud base height, cloud top height, and cloud thickness. Radiosonde measurements are compared with a Ka-band cloud radar at one site and ERA5 reanalysis data. The most interesting finding to me is Figure 7a which shows the global statistics of CBH, CTH, and CT for multi-layer clouds. But, as one of the reviews said, there are several uncertainties associated with the radiosonde measurements and their retrieval method. I have a few minor comments listed below.

Response: We thank the reviewer for his/her comprehensive evaluation and thoughtful comments, which help tremendously to improve the quality of our work. We have tried our best to address the reviewer's concerns one by one. For clarity purpose, here we have listed the reviewer's comments in black, followed by our responses in blue, and the modifications to the manuscript are in italics. We sincerely hope that the reply and the revisions can meet the expectations of editor and reviewer.

**Specific Comments**

1. What is the vertical resolution of ERA5 reanalysis data? "137 levels from the surface to a height of 80 km ASL". For example, what is the resolution at 2 km, 5km, and 10 km. I think ERA5 reanalysis data takes account of the local sounding data. I'm a little surprised to see the large differences between radiosonde and ERA5. Is it because the resolution of ERA5 is too coarse?

Response: The ERA5 reanalysis can provides single level variables (e.g., cloud base height (CBH) and total cloud cover) on one level or near surface, pressure level variables (e.g., temperature and relative humidity) on 37 levels from 1000 to 1 hPa, and model level variables on 137 levels from the surface to a height of 80 km ASL (https://cds.climate.copernicus.eu/cdsapp#!/dataset/reanalysis-era5-complete?tab=overview). We changed the sentence "The product consists of hourly analysis fields on 137 levels, from the surface to a height of 80 km ASL." to "The product consists of *single level variables (e.g., CBH and total cloud fraction) on one level or near surface or one level, pressure level variabl*es (e.g., temperature and relative humidity) on *37 levels from 1000 to 1 hPa, and model level variables on 137 levels from the surface to a height of 80 km ASL (https://cds.climate.copernicus.eu/cdsapp#!/dataset/reanalysis-era5-complete?tab=overview).*" in the revised version (Manuscript_tracked.docx).

Unfortunately, the vertical resolution of CBH does not seem to be documented in ERA5. The CBH in ERA5 is detected using the cloud cover or cloud water mixing ratio threshold. When cloud cover is greater than 1 %, the height from ground to the base is defined as CBH (Wang et al., 2022), which may lead to the underestimation of CBH in ERA5. In addition, the coarse resolution of RH and T profiles from ERA5

reanalysis may be one of the reasons causing the large differences in CBH between radiosonde measurement and ERA5 measurements, as you suggested. Previous studies pointed out that the CBHs of both high-level clouds and relatively low-level clouds were underestimated in ERA5 (e.g., Miao et al., 2019; Wang et al., 2022; Li et al., 2022). The underestimation for high-level clouds in the ERA5 results were caused by the poor specification or parameterization of critical RH from model (Miao et al., 2019). The underestimation of relatively low clouds may be associated with issues of cloud parameterization schemes used for ERA5.

References:

Li, D., Liu, Y. Z., Shao, T. B., Luo, R., and Tan, Z. Y.: Assessment of cloud base height product from ERA5 reanalysis using ground-based observations, Chinese J. Atmospheric Sci., https://doi.org/10.3878/j.issn.1006-9895.2208.22109, 2022. (In press).

Miao, H., Wang, X. C., Liu, Y. M., and Wu, G. X.: An evaluation of cloud vertical structure in three reanalyses against CloudSat/cloud-aerosol lidar and infrared pathfinder satellite observations, Atmos Sci Lett., 20, e906, https://doi.org/10.1002/asl.906, 2019.

Wang, R. J., Zhou, R. J., Yang, S. P., Li, R., Pu, I. P., Liu, K. Y., and Deng, Y.: A new algorithm for estimating low cloud-base height in southwest China, J. Appl. Meteorol. Clim., 61, 1179–1197, https://doi.org/10.1175/JAMC-D-21-0221.1, 2022.

2. Eq.5: I understand the authors want to have an RH profile to determine the cloud layer. RH_liquid and RH_ice both have a physical meaning, that is the relative humidity respect to water and ice, respectively. However, RH_mixed is just a combination of RH_liquid and RH_ice. It is not correct to say "RH_mixed is the RH with respect to liquid-ice mixed".

Response: Thank you very much for pointing out the mistake. We changed the sentence "To obtain the RH profile with respect to liquid-ice mixed ($RH_{mixed}(z)$), …" to "In this case, the RH is termed as $RH_{mixed}$, which is a combination of $RH_{liquid}$ and $RH_{ice}$. To obtain the $RH_{mixed}$ profile, the solutions of ice phase and liquid phase are scaled linearly with T(z) (Austin et al., 2009). …" in the revised version (Manuscript_tracked.docx).

Reference:

Austin, R. T., Heymsfield, A. J., and Stephens, G. L.: Retrieval of ice cloud microphysical parameters using the CloudSat millimeter-wave radar and temperature, J. Geophys. Res.-Atmos., 114, D00A23, https://doi.org/10.1029/2008JD010049, 2009.

3. Eq.7-9: I think it is smart to use the first and second derivatives to find the cloud layers. Is it possible that Eq. 7 is satisfied due to some fluctuations of T and RH in nature? For example, when Eq. 7 is satisfied it might be a false signal, not the true cloud layer.

Response: Thanks for your recognition. Yes, Eq. 7 is satisfied when there are some fluctuations in air temperature (T) and relative humidity (RH) in nature. Thus, only using the vertical gradients of T and RH to detect cloud layers may cause error. To obtain the accurate cloud vertical structures, we combine both the vertical gradients of T and RH and the altitude-dependent thresholds of RH to detect cloud layers. After detecting moist layers by using the first and second derivatives of T and RH, we identified cloud layers by using height-resolving RH thresholds defined in Table 2.

4. Figure 5: Because radiosonde observations are compared with radar at one site, I would recommend the author to plot the same figure but only using data from MMCR. The figure can be in the supplementary. I guess the agreement would be better.

Response: If I understand correctly, you suggest comparing the CBHs and cloud top heights (CTHs) from radiosonde measurements with those from MMCR at 374 radiosonde stations, which is similar to Fig. 5. We also guess the agreement would be better. Unfortunately, we only obtained the cloud vertical structure (CVS) from MMCR at one site (Beijing Nanjiao weather observatory). When MMCR data are available at a mass of sites in the future, we will compare the radiosonde-derived CVS with those from MMCR on a global scale.

---

## Author Response (AR3)

**Response to Reviewer 2' Comments**

Comments on "Characterizing the near-global cloud vertical structures over land using high-resolution radiosonde measurements" by Xu et al.

We thank the reviewer for his/her comprehensive evaluation and thoughtful comments, which help tremendously to improve the quality of our work. We have tried our best to address the reviewer' concerns one by one. For clarity purpose, here we have listed the reviewer' comments in black, followed by our responses in blue, and the modifications to the manuscript are in italics. We sincerely hope that the reply and the revisions can satisfy the editor and referee' expectations.

**Specific Comments:**

1. L69-70: The last sentence appears to be repeated twice here. Remove one of them.
Response: Thanks for pointing out this mistake. We deleted one of the sentences in the revised version (Manuscript_tracked.docx).

2. L204-206: It's probably best to move these sentences into the next paragraph so it flows more smoothly. You can also remove the sentence "The detailed description of this method proceeds as follows:"
Response: Thanks for your great suggestions. We deleted the sentence "The detailed description of this method proceeds as follows:" and moved the sentence "In order to improve the accuracy of cloud detection, we develop a CVS retrieval method by combining the vertical gradients of $RH(z)$ and $T(z)$ and altitude-dependent thresholds of RH as well." into the next paragraph in the revised version (Manuscript_tracked.docx).

3. Section 2.2.3.: Given this subsection is quite short, I think it would flow better to rewrite this as a short paragraph as opposed to a 2-item list as currently presented.
Response: Thanks for your great suggestions. We changed the 2-item list to a short paragraph in the revised version (Manuscript_tracked.docx), as follows:
"To obtain robust cloud structures, the cloud layers determined above have to be further reprocessed. *If the distance between two contiguous cloud layers is less than 300 m, or the min-RH between the continues cloud layers is greater than the corresponding minimum RH threshold (inter-RH) between the consecutive cloud layers (Table 2), these two cloud layers are merged (Zhang et al., 2010).*"

4. L597: This is overstated a bit. These results could be used to validate output from global climate models, as constraints for cloud formation would need to be done inside a parameterization. I also think you should add a sentence in this paragraph stating that these results can also be readily used by the operational weather community given the widespread use of radiosondes in day-to-day forecasting operations.

Response: Thanks for your kindly reminder. Here, we revised the sentence "The results of CVS provide a valuable addition to the study of cloud-radiation-dynamics interactions and have the potential to improve the performance of current climate model." as follows:

"*These results of CVS could be used to validate output from global climate models, as constraints for cloud formation would need to be done inside a parameterization. In addition, these results can also be readily used by the operational weather community given the widespread use of radiosondes in day-to-day forecasting operations.*" in the revised version (Manuscript_tracked.docx).